# VIDEOPANDA: VIDEO PANORAMIC DIFFUSION WITH MULTI-VIEW ATTENTION

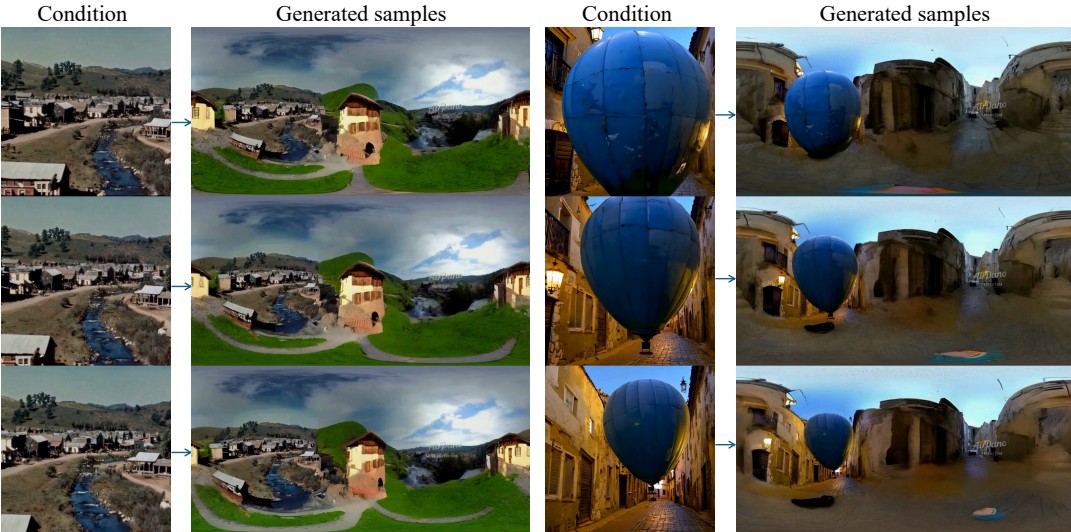

Condition    Generated samples    Condition    Generated samples

*A view of a quaint town nestled between rolling hills, with wooden houses, a winding river, and a clear blue sky overhead.*

*Handheld tracking shot at night, following a dirty blue ballon floating above the ground in abandon old European street.*

Figure 1: Generated samples conditioned on a single-view video and text prompt. Both single-view video inputs were generated using existing video generation models (Brooks et al., 2024; Runway, 2024). Auto-regressive generation is applied to extend the video length.

## ABSTRACT

High resolution panoramic video content is paramount for immersive experiences in Virtual Reality, but is non-trivial to collect as it requires specialized equipment and intricate camera setups. In this work, we introduce VideoPanda, a novel approach for synthesizing $360°$ videos conditioned on text or single-view video data. VideoPanda leverages multi-view attention layers to augment a video diffusion model, enabling it to generate consistent multi-view videos that can be combined into immersive panoramic content. VideoPanda is trained jointly using two conditions: text-only and single-view video, and supports autoregressive generation of long-videos. To overcome the computational burden of multi-view video generation, we randomly subsample the duration and camera views used during training and show that the model is able to gracefully generalize to generating more frames during inference. Extensive evaluations on both real-world and synthetic video datasets demonstrate that VideoPanda generates more realistic and coherent $360°$ panoramas across all input conditions compared to existing methods. Visit the project website at https://mvpanovideo.github.io/VideoPanda/ for results.

## 1 INTRODUCTION

A key aspect of achieving true immersion in a virtual environment is allowing users to look around freely, by rotating their head and exploring their surroundings from all possible angles. To en-

able such experiences, it is essential to have access to high-quality and high-resolution panoramic videos. However, recording such videos is both expensive and time-consuming, as it requires intricate camera setups and specialized equipment. As a result, the available panoramic video content on platforms such as YouTube or Vimeo remains limited compared to single-view videos. In this work, we aim to address this issue, by developing a generative model capable of synthesizing panoramic videos either from text prompts or by expanding single-view videos (either generated from models like Sora (Brooks et al., 2024) or recorded) into panoramic format. We consider this an essential step towards making immersive content more accessible and scalable.

Recently, diffusion models have shown remarkable success in generating images (Ho et al., 2022; Blattmann et al., 2023a), 3D models (Shi et al., 2023b; Poole et al., 2022), and videos (Brooks et al., 2024; Blattmann et al., 2023b) from text prompts. Despite their promising capabilities, generation of panoramic videos using diffusion models presents significant challenges, mainly due to the scarcity of high-quality panoramic video datasets. Furthermore, while substantial progress has been made towards advancing standard video generation pipelines (Girdhar et al., 2023; Hong et al., 2022; Chen et al., 2024; Zheng et al., 2024), very few works have attempted to apply these techniques to panoramic video generation. Existing methods are either limited to specific domains such as driving scenarios (Wen et al., 2024; Wu et al., 2024; Li et al., 2023; Zhao et al., 2024; Liu et al., 2024b) or restricted to generating static scenes (Wu et al., 2023; Zhang et al., 2024). 360DVD (Wang et al., 2024a) directly generates equirectangular panorama video (with text condition), which presents a large domain gap to base model pretrained on perspective view. We perform an extensive comparison to 360DVD in the text-conditional setting and demonstrate our improved visual quality.

In this paper, we introduce **VideoPanda**, a novel approach capable of generating high-quality panoramic videos from text prompts and single-view video, as well as creating long video using auto-regression. Our approach builds on existing video diffusion models by adding multi-view attention layers to generate consistent multi-view outputs. Doing so ensures that the output domain (perspective images) remains close to the original training distribution of the pretrained video model (as opposed to directly generating equirectangular projections), which helps in maintaining video quality while generating multiple views. The resulting views are then seamlessly stitched together to create a cohesive panoramic video. We evaluate our model on a diverse set of data domains, including both real and synthetic videos, and demonstrate its superior performance and quality compared to previous approaches, both quantitatively and qualitatively. Additionally, a user study indicates that the majority of participants prefer our generated videos over those from other baseline models. In summary, we make the following contributions:

- We identify the value of panoramic video generation by allowing users to input single-view videos as a condition – a widely available modality, and present a multi-view video architecture capable of generating plausible panoramic videos.
- We demonstrate that our model can be jointly trained for text-conditioning, video-conditioning, and autoregressive settings by randomizing the conditioning type, leading to improved results and enabling the generation of long panoramic videos.
- When extending the video model to multi-view, the number of generated image frames greatly increases. We overcome the inherent computational burden associated by randomly subselecting the number of views and frames and show that it gracefully generalizes to video with long duration and more views during inference.

## 2 RELATED WORK

### 2.1 IMAGE AND VIDEO DIFFUSION MODELS

Diffusion models (Ho et al., 2020) have demonstrated remarkable success in generating high-quality images (Karras et al., 2022; 2024; Pernias et al., 2023; Hoogeboom et al., 2023; Ho et al., 2022) and videos (Girdhar et al., 2023; Hong et al., 2022; Blattmann et al., 2023a;b; Brooks et al., 2024; Guo et al., 2023; Chen et al., 2023; Gupta et al., 2023) from text prompts. To reduce the computational cost of generating high-dimensional data such as images and videos, latent diffusion models (Rombach et al., 2022) (LDMs) proposed to first encode the data into a compressed latent space using a variational autoencoder (VAE) (Kingma, 2013a), and then conduct the diffusion in this lower dimensional space. These models have been proven highly effective for a wide range of downstream tasks

such as inpainting (Lugmayr et al., 2022), controllable generation (Zhang et al., 2023), customized generation (Ruiz et al., 2023), and image/video editing (Kawar et al., 2023; Molad et al., 2023) etc.

## 2.2 Multi-View Image Generation

Building on the success of diffusion models for 2D image generation, they have been increasingly adapted also for multi-view image generation. However, due to the limited availability of real-world multi-view training data, several recent approaches (Shi et al., 2023b; Long et al., 2024; Liu et al., 2023b;a) attempted to fine-tune pretrained image generation models like Stable Diffusion (Rombach et al., 2022) to support multi-view generation. Such approaches can be roughly categorized into two categories: object-centric and scene-centric approaches.

Object-centric models focus primarily on generating images of objects where all cameras are inward-facing, looking at a single object from different viewing directions. Examples of such approaches include (Kant et al., 2024; Kong et al., 2024; Shi et al., 2023a;b; Tang et al., 2024; Voleti et al., 2024; Wang & Shi, 2023). More recently, several object-centric generative models explored incorporating custom attention mechanisms (Hu et al., 2024; Huang et al., 2023; Kant et al., 2024; Li et al., 2024b) to aggregate view-specific information across multiple views. Notable among these, CAT3D (Gao* et al., 2024) trains a model that generates novel views of an inward-focused scene from one or more input views, allowing for 3D reconstruction from a single image. However, these methods often focus on single-object scenes, which limits their applicability to more complex environments.

The second line of work seeks to generate realistic multi-view images of entire scenes, using outward-facing cameras to capture different viewing directions and produce panoramas. For instance, PanoDiffusion (Wu et al., 2023) is trained on equirectangular projections of 360° panoramic images, and relies on inpainting during inference to extend the input images into complete panoramas. Building on this, PanFusion (Zhang et al., 2024) adds an additional branch to Stable Diffusion, enabling the simultaneous generation of panoramas and multi-view images. MVDiffusion (Tang et al., 2023) introduces correspondence-aware attention (CAA) layers, where each point attends only to other points within its local neighborhood. More recently (Yuan et al., 2024; Wang et al., 2023) proposed predicting the homography between input views and use a diffusion model to generate the unseen regions of the panorama. Lastly, LayerPano3D (Yang et al., 2024) combines multi-view and inpainting models to generate multi-layer panoramas, allowing for somewhat limited exploration within the scene boundaries. Other notable works in this area include (Li et al., 2024a; Zhou et al., 2024; Hara & Harada, 2024; Liu et al., 2024a).

## 2.3 Multi-view and Panorama Video Generation

The emergence of powerful open-source video diffusion models (Blattmann et al., 2023a; Chen et al., 2024; Zheng et al., 2024) gave rise to the development of several approaches aimed at augmenting them with multi-view capabilities (Watson et al., 2024) and extending them to generate 360° panoramic videos. For example, 360DVD (Wang et al., 2024a) builds upon a pretrained text-to-video model (Guo et al., 2023) by adding a 360-adapter and fine-tuning it on equirectangular projections of panoramic videos. This enables the creation of 360° videos from a text inputs, with the option to condition on optical flow videos. Generative Camera Dolly (Van Hoorick et al., 2024) extends the image-conditional Stable Video Diffusion (Blattmann et al., 2023a) into a video-to-video model. Given an input video of a scene, (Blattmann et al., 2023a) can generate a synchronized video from a different camera trajectory. 4K4DGEN (Li et al., 2024c) draws inspiration from MultiDiffusion (Müller et al., 2024) and introduces a training-free method that denoises multiple views of a spherical panorama simultaneously. Most similar to our method is Panacea (Wen et al., 2024), which is inspired byVideoLDM (Blattmann et al., 2023a) and extends StableDiffusion by adding multi-view and temporal attention layers, trained on multi-view driving videos. Notably, Panacea relies on a dynamic birds' eye view (BEV) representation as conditioning, which is most commonly available in the case of driving scenes, thus effectively limiting its applications to driving scenes.

## 3 Method

In this work, we introduce VideoPanda, a multi-view video diffusion model capable of generating long panoramic 360° videos from a text prompt or a perspective video. Below, we describe our

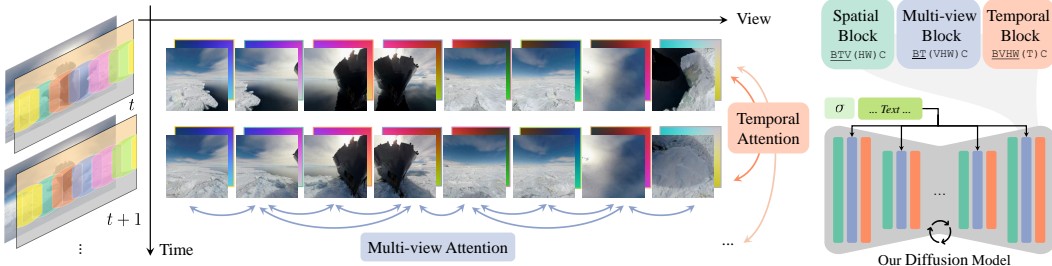

Figure 2: We divide the equi-rectangular video into 8 perspective views via projection. Our diffusion model consists of interleaved spatial, multi-view, and temporal blocks, conditioned on text prompts. Attention is used to propagate information through the multi-view videos to ensure consistency. The input views are embedded using the ray directions as visualized by the color map behind the perspective images.

multi-view video diffusion model (§ 3.1), detail the model training strategy (§ 3.2), and finally describe the approach for auto-regressively generating long videos (§ 3.3). Fig. 2 provides an overview of our general model design.

## 3.1 MODEL DESIGN

We train a multi-view video diffusion model that, given a text prompt and an optional set of conditioning frames, is able to jointly generate multiple multi-view consistent videos of different view directions that together cover a full 360° panoramic video.

Our architecture builds on video latent diffusion models (VLDM) (Blattmann et al., 2023b) by incorporating multi-view attention layers inspired by MVDream (Shi et al., 2023b) and injecting view direction embeddings into the model. Specifically, we add 3D multi-view self-attention layers that perform self-attention across images from different views at each frame of the video. These layers are combined with the existing 2D self-attention layers in a residual manner using zero-initialized convolutions, similar to ControlNets (Zhang et al., 2023). To provide the model with an understanding of viewing directions, we use ray direction representations that are the same height and width as the latent representations and encode the ray directions at each spatial location, following (Gao* et al., 2024). These rays are defined relative to the camera pose of the first view, and are invariant to global 3D translations and rotations. The view embeddings are concatenated channel-wise with their corresponding latents and are fed into the model at the first layer using zero-initialized convolutions.

Given a set of target and optional conditioning frames of size $512 \times 512 \times 3$, each image is encoded into a latent representation of size $64 \times 64 \times 4$ using a variational autoencoder (VAE) (Kingma, 2013b). To enable conditioning on specific frames, we adopt the approach from CAT3D (Gao* et al., 2024). During training, the latents corresponding to the non-conditioned views are noised according to the diffusion process, while the latents of the conditioning frames are kept mostly clean. Following prior work (Ho et al., 2021), to improve robustness and prevent overfitting, we use **noise augmentation** by adding a small amount of noise $\sigma$ to the input conditioning latents and pass this value $\sigma$ to the model as well. A binary mask is concatenated channel-wise to distinguish between the input conditioning latents and the target frames to be predicted. The diffusion model is then trained to learn the joint distribution of these latent representations conditioned on the inputs. We incorporate classifier-free guidance (CFG) (Ho & Salimans, 2022) by randomly dropping the conditioning frames with a probability of 10% during training.

Finally, similar to prior works (Hoogeboom et al., 2023), we observe improved performance when shifting the noise schedule towards higher noise levels, as our model generates more image frames than the base video model. Please see Appendix A.2 for more details. We also find that using a $v$-prediction objective (Salimans & Ho, 2022) leads to more stable training compared to $\epsilon$-prediction, particularly with high-noise schedules.

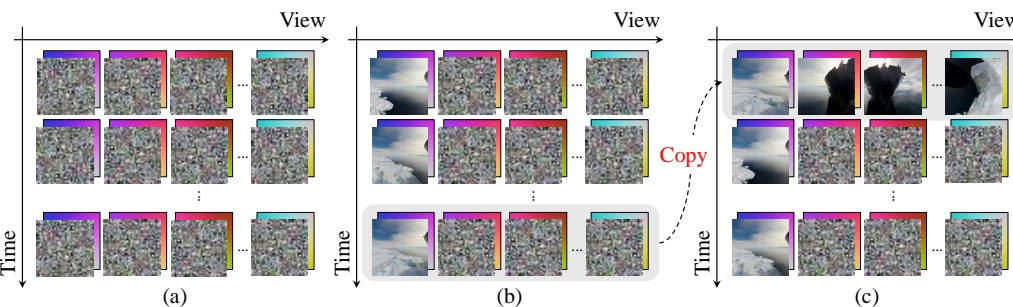

Figure 3: The model is trained using three frame conditioning regimes. (a) No image conditions and the initial inputs are pure noise; (b) Conditioning only on the first view of the video; (c) Conditioning on the first frame and first views for auto-regressive video generation. At inference time, we autoregressively condition on long videos by using conditioning (b) to generate the first window and subsequently using the last multi-view images row from the previous time step (the shaded region) as the first row input to our model using condition-type (c).

## 3.2 TRAINING STRATEGY

We initialize the model from a pretrained text-to-video diffusion model (Blattmann et al., 2023b) which has been trained on web-scale video data. Following prior works (Shi et al., 2023b), the weights of the multi-view attention layers are initialized with the same weights as the existing 2D self-attention layers to accelerate training.

As we want to adjust the noise schedule (shifting toward higher noise levels) and change the model parametrization from $\epsilon$-prediction to $v$-prediction without overfitting the model to our limited panorama videos, we train our model in two stages. In the first stage, we finetune the single-view text-to-video model from the existing checkpoint, adapting it to the new noise schedule and loss objective. This stage is performed on a subset of the original pretraining data with standard captioned videos of 16 frames and requires minimal training time, as the model adapts quickly to these changes. In the second stage, we freeze the spatial layers of the video model and finetune the rest using multi-view video data.

During training, we randomize both the number of views and video frames to enhance the model's generalization and prevent overfitting to the limited $360°$ video data, effectively using this as a form of data augmentation. The model is trained to generate multi-view video sequences represented as view-frame matrices of varying sizes, such as $3 \times 16$, $4 \times 12$, $6 \times 8$, and $8 \times 6$, where the first dimension refers to the number of views and the second to the number of frames. We refer to this randomization as **random matrix** going forward. This allows the model to generalize to new view-frame combinations, like $8 \times 16$ matrices, during inference—configurations that couldn't fit in GPU memory during training.

To handle multiple conditioning scenarios, we train a single general model that can generate multi-view videos conditioned on text, video, or a combination of video and the first frame's multi-view images for autoregressive generation using a **multi-task** training strategy. Specifically, the binary mask is randomized to reflect these different conditioning setups: all zeros (text conditioning), the first column of ones with zeros elsewhere (video conditioning), or both the first row and first column set to ones (autoregressive generation), with equal probability. See Fig. 3 for a visualization of the different types of conditioning.

## 3.3 AUTOREGRESSIVE GENERATION OF LONG VIDEOS

To generate long panoramic videos, we use an autoregressive approach (see Fig. 3). Initially, conditioned on the first 16-frames of the input video, the model generates an $8 \times 16$ view-frame matrix. For subsequent frames, the model is conditioned on the next 15 new frames of the video (a column) and the last frame from all 8 views (a row) generated in the previous step. This iterative process allows us to generate long, coherent video sequences with smooth transitions and consistent motion.

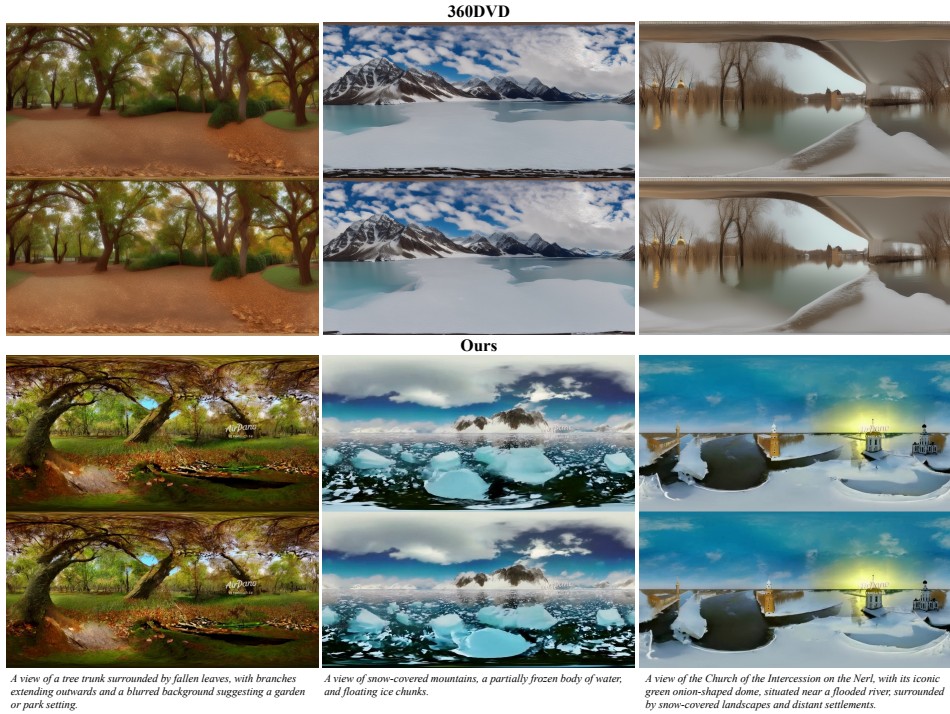

360DVD

Ours

*A view of a tree trunk surrounded by fallen leaves, with branches extending outwards and a blurred background suggesting a garden or park setting.*

*A view of snow-covered mountains, a partially frozen body of water, and floating ice chunks.*

*A view of the Church of the Intercession on the Nerl, with its iconic green onion-shaped dome, situated near a flooded river, surrounded by snow-covered landscapes and distant settlements.*

Figure 4: Qualitative figure compare text conditional video generation, 360DVD VS ours. The pixel quality of 360DVD is lower and distortion near the poles (top and bottom) is worse.

Autoregressive generation, however, tends to accumulate errors over time, leading to a gradual degradation in image quality and noticeable blurring after a few iterations. The noise augmentation introduced in § 3.1 helps mitigate this issue, consistent with findings from prior work (Valevski et al., 2024). This noise augmentation serves two purposes: it acts as a data augmentation technique to improve generalization, and it allows the model to self-correct by learning to recover clean information from noisy samples generated in previous iterations. Please see Appendix A.3 for details.

## 4 EXPERIMENTS

In this section, we explain the details of our experimental setting and our methodology for evaluations. We then present qualitative and quantitative comparisons to assess our models efficacy against baselines in text and video-conditional generation, demonstrate our models extension to long video generation and ablate key components of our training strategy. Additional training details are included in Appendix C.

### 4.1 DATA

**Training Data.** We train our model on the WEB360 (Wang et al., 2024a) dataset, which contains 2,114 panorama video clips with automatically generated captions. Each clip is 100 frames in length, totalling approximately 3 hours of footage that predominantly features panning shots of outdoor scenery.

**Evaluation Data.** For the video conditioning task, we evaluate our method using two sources of data:

- In-distribution condition input: We gather 100 unseen panorama video clips from Youtube and extract 90 FOV horizontal perspective views for the input conditioning. Prompts are obtained by captioning the middle frame of the conditioning video using CogVLM (Wang et al., 2024b).

| MV-Diffusion | Ours |
|---|---|

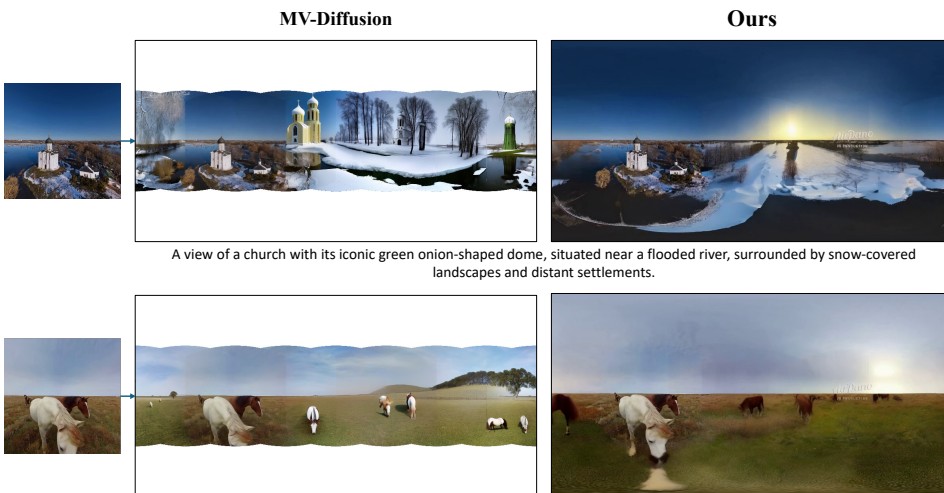

A view of a church with its iconic green onion-shaped dome, situated near a flooded river, surrounded by snow-covered landscapes and distant settlements.

A view of a vast grassy field horses grazing.

Figure 5: Qualitative figure comparing video conditional generation, MVDiffusion VS ours. Note that MVDiffusion can only outpaint each frame of the video separately. MVDiffusion is worse at maintaining the structure and style of the input view globally compared to ours. For example the sky color and the scales and depths of objects is less consistent for MVDiffusion.

- Out-of-distribution condition input: We use generated videos from models including SORA (Brooks et al., 2024), Runway (Runway, 2024) and Luma (Lumalabs, 2024). These videos are cropped and resized to a resolution of $512 \times 512$ and treated as horizontal side views. When available, we use the original prompt; otherwise, we caption the middle frame with CogVLM.

Since the out-of-distribution condition inputs do not originate from 360 videos, we cannot compute metrics that require ground truth images, such as pairwise FVD (Unterthiner et al., 2018) and the reconstruction metrics. Instead, we only evaluate them qualitatively. For the text conditional task, we use two data source: 100 prompts which are derived from captioning the in-distribution videos and 30 prompts generated by ChatGPT.

**Processing.** We first convert the equirectangular video data into multiple perspective views with overlap. A visualization is shown in Fig. 6. Similar to MVDiffusion we cover the horizontal side views with multiple perspective views of 90° FOV at 0° elevation. We empirically observed that the excessive amount of overlap stemming from the use of 8 horizontal views was unnecessary and thus we only use 6 views instead. These are evenly spaced in azimuth in offsets of 60°. We also explored using just 4 views which results in no overlaps between views similar to a cubemap representation but found that it was more difficult for the model to maintain consistency between views without overlaps. Additionally, to obtain a full panorama we add two perspective views looking straight up (90° elevation) and down (-90° elevation) to cover the 'sky' and 'ground' views. We increase the FOV for these two to 100° which is large enough to cover all pixels in the panorama when combined with the 6 side views.

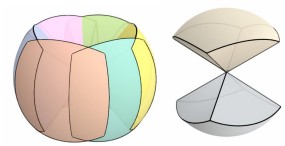

Figure 6: A visualization of the 8 frames used during training, consisting of 6 horizontal views with 90 FOV and 2 views for the top/bottom with 100 FOV

### 4.2 INFERENCE

Unless otherwise specified, we use a DDIM sampler with 25 steps and classifier-free guidance (CFG) to improve generation quality. In the text-conditional setting, we use a CFG scale of 8.0. For the video-conditional setting, we use a CFG scale of 4.0, where the unconditional score prediction does not take text nor video as input.

To facilitate fair evaluation, we use a common equirectangular format with resolution $512 \times 1024$ for a 16 frame long panoramic video and compose our multi-view results into it by warping each of the images with bicubic interpolation. The pixel values in regions with overlap between views are uniformly averaged. When evaluating our generations, we either directly evaluate the stitched equirectangular video or following MVDiffusion, we crop 8 horizontal perspective view videos from it, since some metrics are more naturally evaluated using perspective views as input.

### 4.3 METRICS

**Validation Pair FID and FVD.** On validation sets we compare the set of generated frames to their paired real unseen frames in aggregate distribution. This evaluates both the quality and favors generations that adhere more closely to the true frames.

**Reconstruction Metrics.** In the video-conditional setting, we directly compare generated frames to their real counterparts as is commonly done for evaluating novel view synthesis performance. We use PSNR, SSIM and LPIPS (Zhang et al., 2018). Note that evaluating reconstruction metrics in the conditional generative setting can be problematic as the desired output is inherently ambiguous. Namely, direct comparisons with the ground truth can favor mode covering solutions, that may be lower in diversity.

**Clip Score (Clip).** We evaluate alignment to the supplied text prompt via clip score.

**User Preference.** We additionally conduct a user study, where equirectangular video/images from our model and the baseline are shown side-by-side to the user along with the conditioning input and they are asked to select their preferred result. For this setting, we randomly subsample 20 videos for each comparison and conducted the study with 6 users.

### 4.4 TEXT-CONDITIONAL GENERATION

We evaluate our model's ability to generate multi-view videos from a text prompt and compare it to 360DVD (Wang et al., 2024a) that is our primary baseline. A quantitative comparison is summarized in Tab. 1. We note that our model outperforms 360DVD across all metrics. A side-by-side visual comparison is provided in Fig. 4, demonstrating that VideoPanda produces videos with higher image quality and sharper details. In contrast, 360DVD's outputs extremely blurry and undersaturated results that suffer from insufficient warping near the top and bottom of the panoramas, hence leading to noticeable stretching artifacts when viewed in 3D, as we show in Appendix Fig. A5.

| | Panorama | | | Horizontal 8 views | | | User |
|---|---|---|---|---|---|---|---|
| | $FID_{pair} \downarrow$ | $FVD_{pair} \downarrow$ | Clip $\uparrow$ | $FID_{pair} \downarrow$ | $FVD_{pair} \downarrow$ | Clip $\uparrow$ | Pref$\uparrow$ |
| 360DVD | 160 | 1942 | 28.4 | 128.7 | 958.2 | 27.6 | 28% |
| Ours (multi-task) | **136** | **1258** | **29.8** | **91.3** | **600.5** | **28.9** | **72%** |

Table 1: Quantitative comparison for text-conditional panorama video generation.

### 4.5 VIDEO-CONDITIONAL GENERATION

Our video-conditional model accepts both a single view video and a text prompt which can be obtained through captioning the input view. During training, we randomly select one of the horizontal views, as shown in Fig. 6, as the conditional one and do not apply any noise on it. We exclude conditioning on top and bottom views as this case is less common. During inference we directly treat the input video as one of our horizontal views.

For general videos, there are no existing models that consider the video-conditional panoramic video generation task. Therefore, we compare our model to existing image-conditional panorama image generation model, MVDiffusion, at the frame level. In particular, for our method, we first generate a 16 frame panorama video and then extract the middle frame. We compare against the outpainting model from MVDiffusion and report the results in Tab. 2. Since MVDiffusion does not cover the sky or ground regions, we only evaluate metrics on the 8 horizontal views. Our method scores significantly better on FID and reconstruction metrics, while being slightly worse on the clip score. Qualitatively we find that our method is much better at maintaining the style and scene scale/depth

| | Horizontal 8 views | | | | | User |
|---|---|---|---|---|---|---|
| | FID ↓ | Clip ↑ | PSNR ↑ | LPIPS ↓ | SSIM ↑ | Pref↑ |
| MVDiffusion | 96.8 | **29.7** | 13.4 | 0.568 | 0.485 | 23% |
| Ours (multi-task) | **63.2** | 28.5 | **17.6** | **0.457** | **0.636** | **77%** |

Table 2: Quantitative comparison of single view video-conditional panorama generation with image panorama outpainting method MVDiffusion. We extract the middle frame from our 16 frame generations to compare at a per image level.

| Ours Ablation | | Panorama | | | | Horizontal 8 view videos | | | |
|---|---|---|---|---|---|---|---|---|---|
| multi-task | rand-mat | FID↓ | FVD↓ | Clip ↑ | PSNR ↑ | FID ↓ | FVD ↓ | Clip ↑ | PSNR ↑ |
| ✓ | ✓ | **98** | 916 | **29.6** | 15.9 | 49.8 | 258 | **28.6** | 17.6 |
| × | ✓ | 103 | **861** | 28.9 | 16.0 | **48.4** | **255** | 28.2 | 17.3 |
| × | × | 124 | 999 | 27.1 | **17.0** | 69.8 | 445 | 26.0 | **18.5** |

Table 3: Quantitative ablations of our model on single view video-conditional panoramic video generation. Training our model to be multi-task capable incurs a negligible drop in performance. Randomizing the matrix of frames during training results in much improved video quality at a slightly worse color consistency as measured by PSNR.

in the other generated views as demonstrated in the qualitative examples from Fig. 5. We also tried comparing to PanoDiffusion but found that this model is prone to over-fitting to indoor room scenes. We additionally, perform video-conditional generation on out of distribution videos and show generated results in Fig. 1 and our project website.

## 4.6 AUTOREGRESSIVE GENERATION

To demonstrate our model's performance on long video generation, we run 4 iterations of autoregression, resulting in a total of $4 \times 15 + 1 = 61$ frames for the panorama videos. We observe that, despite using noise augmentation, autoregressive errors gradually accumulate, causing the scene to become blurry. To mitigate this, the noise-augmentation value can be increased during inference to regenerate finer details, though this introduces slight flickering due to the newly added details. Ideally, a dynamic system could be developed to increase the value when blurriness occurs and reduce it otherwise, minimizing flickering while keeping pixel quality high—an avenue we leave for future work. We provide examples of extracted frames from our autoregressively generated videos in Fig. 1 and Fig. B1. Please see our website for best viewing of long video generations.

## 4.7 ABLATIONS

We ablate the main components of our method and include additinal ablations on shifting the noise schedule of the base model, the architecture for conditioning on image frames and noise augmentation in Appendix A.

**Random Matrix vs. Fixed Matrix.** During training, we can fit a maximum of 6 time frames with 8 multi-views in memory. However, at inference we wish to generate 16 frames which is the native frame length for our base video model and aligns with 360DVD. To enable this we employ the randomized matrix strategy described in § 3.3. To evaluate the benefit of this strategy, we compare $8 \times 16$ video-conditional generations from a model that was trained with the "random-matrix" strategy using an even mix of $8 \times 6$, $4 \times 12$ and $3 \times 16$ with one using a "fixed-matrix" strategy trained with only the $8 \times 6$ setting. We include a quantitative comparison in Tab. 3 and qualitative examples in Fig. 7. From the comparison, we see that the fixed-matrix trained model can create more blurry regions in its generations which is also reflected in significantly higher FID and FVD and somewhat lower clip score. Reconstruction metrics are very similar but slightly prefer the fixed-matrix model. We hypothesize that focusing training more on the 8 view case could slightly improve the global color consistency at the cost of worse visual quality.

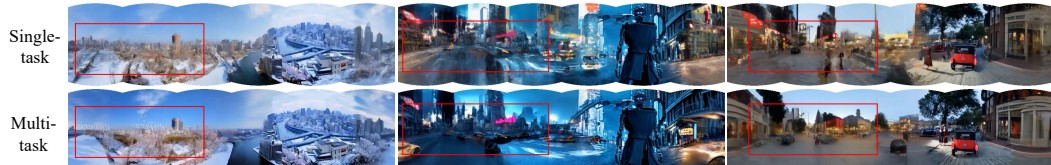

Figure 7: Qualitative figure comparing full matrix and random matrix training. Random matrix training generates more high frequency details.

Figure 8: Qualitative figure comparing our single task vs multi-task model both generating 6 views on out of distribution video. Multi-task training provides better pixel quality. Moreover, with multi-task training, we can train one unified model for different tasks including video conditional generation and auto-regressive generation.

**Multi-task Training.** We find that we can train one unified model to handle text-only conditioning, single view video conditioning and autoregressive conditioning. In Tab. 3 we also quantitatively compare our multi-task model with one only trained for the video conditional setting. For all the metrics, the multi-task model is marginally worse but very close indicating that we can train our model jointly with negligible impact to the quality. We also observe on some OOD conditions, that the random conditioned model tends to improve pixel quality slightly as seen in Fig. 8 which could be due to better generalization from multi-task training.

## 5 CONCLUSION

We present VideoPanda, a model for panoramaic video generation. VideoPanda augments a pre-trained video diffusion model with the ability to generate consistent multiview videos that together cover a full panoramic video. We train VideoPanda in a unified manner with flexible conditioning supporting text and single-view video-conditioning and further support auto-regressive generation of longer videos.

Although VideoPanda demonstrates compelling results, there is still room for further improvement. The generation capabilities of our model are restricted by the performance of the base video model and further improvements could be obtained by applying these techniques to more powerful video diffusion models. Our model currently requires the field of view and elevation of the conditioning input to be sufficiently close to the configuration used in training. This could be addressed by estimating these parameters as demonstrated by recent work in the image generation domain (Yuan et al., 2024). Our autoregressive generation balances a trade-off between maintaining image quality over time and consistency between windows which motivates investigating methods that could efficiently achieve both.

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
