# OpenReview forum: "VideoPanda: Video Panoramic Diffusion With Multi-view Attention"
_ICLR.cc/2025/Conference — Submitted to ICLR 2025_

### Official Review · Reviewer_U9nk · 2024-10-28

**Soundness:** 3
**Presentation:** 2
**Contribution:** 2
**Rating:** 5
**Confidence:** 5

**Summary:**

This paper proposes a framework for generating panoramic videos, based on a pretrained video generation model. The authors introduce multi-view generation capabilities by embedding several Multi-view Blocks. Additionally, they propose a random matrix training strategy, using videos of random frames and views to train the model, which increases the model's generalization to longer/more viewed videos while conserving computational resources. The authors trained the model using WEB360 and tested their method on 100 in-distribution and 100 out-of-distribution videos.

**Strengths:**

1. The authors successfully adapted a pretrained video generation model to the multi-view/panoramic video generation task, achieving relatively good visual results.
2. The proposed method supports using text or video as conditions to generate panoramic videos.

**Weaknesses:**

1. In terms of model design, the authors introduced a Multi-view Block on top of SVD (Stable Video Diffusion) to enable multi-view generation, a concept similar to MVDiffusion. However, while MVDiffusion focuses on panoramic image generation, this paper is designed for panoramic video generation, adding a temporal block (see Figure 2). Essentially, this paper can be seen as an application combining SVD and MVDiffusion, with the spatial and temporal blocks derived from SVD and the multi-view block from MVDiffusion. From this perspective, the novelty of the proposed method may be somewhat limited, so it would be beneficial for the authors to explain how their approach differs from these methods and whether they have made specific design choices to address challenges unique to this application.
2. Regarding the training strategy, the proposed random matrix strategy is essentially a compromise due to limited computational resources; theoretically, using more views or frames in training would yield better results. From the experimental results in Table 3, it can be seen that improvements in FID and FVD scores are achieved at the cost of PSNR (rows 2 and 3).
As for the multi-task strategy proposed by the authors—randomly dropping some conditions (such as text, the first frame, or single-view video), or conditioning on only part of the information or a subset of modalities—is a common trick in training diffusion models. For example, dropping text is often used in text-to-image/video tasks [GLIDE, SVD, Imagen, etc.], or conditioning on text and image in image editing [InstructPix2Pix, Tim Brooks et al. 2022]. Therefore, it would be helpful for the authors to clarify how their approach differs from these methods to demonstrate the novelty of their method.

**Questions:**

1. In Table 3, after applying the multi-task training strategy, several metrics, such as FVD and PSNR, show a decline. Could the authors provide an appropriate explanation for this?
2. In section 4.6, the authors tested autoregressive long video generation, producing videos up to 61 frames. Have the authors attempted to generate even longer videos? As video length increases, does generation quality continuously degrade? If so, what feasible strategies might help mitigate this issue?
3. In Table 1, the authors compare their method with 360DVD and report performance improvements. However, 360DVD uses the text-to-image model SD1.5, while the proposed method uses the SVD video generation model, which inherently offers an advantage in video smoothness. Did the authors conduct a relatively fair comparison, such as applying SVD to 360DVD or using SD1.5 with the proposed method?

---

> ### Author Response · Authors · 2024-11-25
> **Response to reviewer U9nk Part1**
>
> We thank the reviewer for their insightful comments and suggestions.
> We would also like to point the reviewer to the additional results and evaluations we have prepared for the rebuttal and presented in the summary response to all reviewers above including:
>
> 1 Expanding and Improving OOD Evaluations
>
> 2 Results on new base model (CogVideoX-2B)
>
> 7 OOD Evaluation: Quantitative Metrics
>
> Updated qualitative results can be seen on our updated webpages below:
>
>
> OOD Text-cond VBench prompts: https://mvpanovideo.github.io/VideoPanda/release_text_cond_comp/release_text_cond_comp/index.html
>
> OOD Col-cond AI generated videos: https://mvpanovideo.github.io/VideoPanda/release_col_cond_comp/release_col_cond_comp/index.html
>
>
> We address each of the points mentioned in the review below:
>
>
> > **Re: Weakness1 “Essentially, this paper can be seen as an application combining SVD and MVDiffusion, with the spatial and temporal blocks derived from SVD and the multi-view block from MVDiffusion.” “it would be beneficial for the authors to explain how their approach differs from these methods and whether they have made specific design choices to address challenges unique to this application”**
>
> Please see point 5 in our response to all reviewers in which we extensively comment about the novelty of our work and relate them to similar works.
>
> We would also like to highlight our multi-task training strategy for letting the same model handle flexible conditioning (see point 5c) and the randomized matrix strategy that shows we can greatly improve the quality of test time generalization to longer sequences with less computational demands (see point 5d).
>
>
>
> > **Re: Weakness1 “Regarding the training strategy, the proposed random matrix strategy is essentially a compromise due to limited computational resources;  theoretically, using more views or frames in training would yield better results. From the experimental results in Table 3, it can be seen that improvements in FID and FVD scores are achieved at the cost of PSNR (rows 2 and 3)”**
>
> (Please also see point 5D in our response to all reviewers)
>
> We agree with the reviewer, as we state in the paper, line 476 the random matrix strategy is a technique to be able to handle larger time horizons for a given compute constraint. This improves the test-time generalization we observe from our model being able to handle longer time horizons.
> PSNR decreases a bit but it is also not a perfect metric for the generative setting where the solution space is inherently ambiguous and a mode covering solution may obtain a better PSNR than a plausible looking generative distribution of samples. We hypothesize based on the qualitative comparison result in Figure 7 in the paper that the effect of random matrix is to greatly improve the visual fidelity of the generation (bottom row) whereas fixed matrix training when extended in horizon at test time generates oversmooth content in the unseen views (top row). This is also consistent with the improvements in FID and FVD, we see.
>
>
> > **Re: Weakness2 and Question1 (Multi-task strategy) “As for the multi-task strategy [...]  it would be helpful for the authors to clarify how their approach differs from these methods to demonstrate the novelty of their method”, “In Table 3, after applying the multi-task training strategy, several metrics, such as FVD and PSNR, show a decline. Could the authors provide an appropriate explanation for this?”**
>
> (Please also see point 5C in our response to all reviewers)
>
> For multi-task training, the purpose is to enable the same model to flexibly handle text, image and autoregressive generation. Such a model has more capabilities than any single model and is more expedient than training a separate model for each setting. Quality wise the metrics in Table 3 do decline a bit but they remain very similar. Indicating that we can use one model for all tasks while retaining a similar level of quality on an individual task.
>
>
> **>Re: Question2 “In section 4.6, the authors tested autoregressive long video generation, producing videos up to 61 frames. Have the authors attempted to generate even longer videos? As video length increases, does generation quality continuously degrade? If so, what feasible strategies might help mitigate this issue?”**
>
> Thanks for the suggestion and it is an interesting exploration to consider. We are working on this result and will share it shortly!

---

> ### Author Response · Authors · 2024-11-25
> **Response to reviewer U9nk Part2**
>
> >**Re: Question3 “However, 360DVD uses the text-to-image model SD1.5, while the proposed method uses the SVD video generation model, which inherently offers an advantage in video smoothness. Did the authors conduct a relatively fair comparison, such as applying SVD to 360DVD or using SD1.5 with the proposed method?”**
>
> (Please see point 2&3 in our response to all reviewers)
>
> To clarify we did not use SVD as the base model for our experiments.
> In the main paper we used the VideoLDM model (not SVD) which was presented in the work Align your Latents [Blattmann et al., 2023a)] as cited in our paper. This model is based on SD1.5 and is quite similar in design to AnimateDiff.
>
> One main benefit of using multiview perspective generation compared to the equirectangular format used by 360dvd, is that we can more naturally generate content in the sky and ground views (above or below +/-60 degree elevation) that typically are highly distorted in the equirectangular format. We show this in the appendix contained in the supplementary material (please see Appendix Figure A5). It is also very visible in the OOD text-conditional samples when viewing the sky or ground views using the provided VR viewer: https://mvpanovideo.github.io/VideoPanda/release_text_cond_comp/release_text_cond_comp/index.html

---

> ### Comment · Reviewer_U9nk · 2024-11-27
>
> Thank you to the authors for their detailed response. The newly added experiments, including the use of a stronger video generation model (CogVideoX) and training with more data, have indeed improved the quality of panoramic video generation. However, my two primary concerns remain unresolved:
>
> 1. **Method Novelty**: The authors acknowledge that their multi-view attention is essentially the same as that in MVDream, arguing that applying it to panoramic video generation constitutes novelty. However, in my view, generating panoramic videos is very similar to multi-view video generation. A single frame of a panoramic video is essentially a composite of images from multiple views, making it fundamentally a multi-view image.
>
> 2. **Fair Comparisons**: As Reviewer cZZr also pointed out, the authors use more powerful video generation models like VideoLDM/CogVideoX as their base, whereas the comparison method (360DVD) uses the image-generation model SD1.5. This inherently disadvantages 360DVD in terms of temporal consistency and motion smoothness. As such, the comparison is relatively unfair, making it difficult to determine whether the performance gains are due to the proposed method itself or the superior capability of the base generation model.

---

> > ### Author Response · Authors · 2024-11-27
> > **Clarifying our comparison with 360DVD using fair data and model setting. Reply to: Official Comment by Reviewer U9nk**
> >
> > We thank the reviewer for their quick reply and kind comments about our updated results and presentation! We very much appreciate your constructive feedback. See below our reply which is also similar to the reply to Reviewer cZZr.
> >
> >
> > We would like to clarify a few points. In our expanded OOD evaluations, we have made a fair comparison (in terms of data and base model) to 360DVD using the **original model from our paper**.
> >
> > All of our quantitative and qualitative comparisons against 360DVD include our VideoPanda model based on **VideoLDM using the same data (WEB360)** which is a fair base model comparison. **VideoLDM is based on SD1.5** and shares a very similar structure to AnimateDiff (the base model of 360DVD).
> >
> > **For quantitative comparisons:**
> >
> > We conducted quantitative OOD text-conditional evaluation where our base model using VideoLDM strongly outperforms 360DVD in Clip Score, FID and FVD (to HDVila3k). We have copied the quantitative comparison here for your convenience:
> > | Text conditional  Vbench All Dimensions (946 prompts x 3 seeds) | Elevation=+/-60degree 8 Views |                         |             | Horizontal 8 Views (elevation=0) |                         |             |
> > |-----------------------------------------------------------------|:-----------------------------:|:-----------------------:|:-----------:|:--------------------------------:|:-----------------------:|:-----------:|
> > |                                                                 | FID (to HDVila3k frames)      | FVD (to HDVila3k video) | Clip Score  | FID (to HDVila3k frames)         | FVD (to HDVila3k video) | Clip Score  |
> > | 360DVD (AnimateDiff + web360)                                          | 149.7                         | 901.1                   | 23.39       | 127.1                            | 801.9                   | 27.63       |
> > | Ours (VideoLDM + web360)                                        | **130.6**                     | **826.6**               | **24.12**   | **112.2**                        | **677.8**               | **27.78**   |
> >
> > These additional comparisons corroborate our findings in the paper on the in-distribution text-conditional setting (Table 1) where our method also greatly outperforms 360DVD.
> >
> >
> > **For qualitative comparisons:**  (https://mvpanovideo.github.io/VideoPanda/release_text_cond_comp/release_text_cond_comp/index.html  )
> >
> >
> > The left column is our VideoPanda with (VideoLDM+WEB360) and the right column is 360DVD (AnimateDiff+WEB360).
> >
> > **Multiview perspective generation VS direct equirectangular generation: **
> > One main benefit of using multiview perspective generation compared to the equirectangular format used by 360dvd, is that we can more naturally generate content in the sky and ground views (above or below +/-60 degree elevation) that typically are highly distorted in the equirectangular format. This is very visible in the OOD text-conditional samples from 360DVD when viewing the sky or ground views using the provided VR viewer. We also showed this in the appendix contained in the supplementary material (please see Appendix Figure A5).
> >
> > Viewing results in VR also highlights the strange motions in 360DVD including the ground morphing in a spiral direction (0) “In a charming Parisian café, a panda”
> > or inanimate objects sliding in different directions (17) “A serene scene unfolds with a delicate porcelain teacup”.
> > Such errors are hard to visually spot from only looking at the equirectangular projection native to 360DVD but very important for the actual task of panorama video generation.
> >
> >
> > **Superior range of capability:**
> >
> > Additionally, 360DVD only considers the text-conditional case and doesn't consider video-conditional or autoregressive generation at all. We can handle all cases with the same base model. I think this can also be seen as a superiority in the range of capability.
> >
> >
> > We would like to reiterate that our contributions are introducing the video-conditional panorama generation. Our multi-task training strategy enables a unified model capable of flexible conditioning during inference  (see point 5c) and the randomized matrix strategy improves the quality of test time generalization to more frames with less computational demands in training (see point 5d).
> >
> >
> > Please also note that for all of the text-conditional comparison to 360DVD **we did not increase the training data**, but kept it the same (WEB360) for fair comparison. Specifically, both the left column VideoPanda with (VideoLDM+WEB360) and middle column VideoPanda with (CogVideoX-2B+WEB360) are using WEB360 dataset.
> >
> >
> > Again we are very very thankful for your prompt and valuable feedback and please let us know if you still have concerns and we will be happy to actively address them!

---

### Official Review · Reviewer_EySz · 2024-11-02

**Soundness:** 3
**Presentation:** 3
**Contribution:** 2
**Rating:** 6
**Confidence:** 3

**Summary:**

This work present a method for generating videos from text or single-view video data, which employs multi-view attention to enhance a video diffusion model and produce consistent multi-view content suitable for immersive panoramas. The work demonstrates performance improvements and provides code.

**Strengths:**

1. The ideas are very easy to understand, reliable and well written.

2. The boost in quantitative metrics looks good.

3. Open source code for easy reproduction by readers

**Weaknesses:**

1.Poor qualitative results. I feel that the overfitting is evident in the effect, that is, the watermarks are being generated, and on closer inspection you can see the airpano.

2.Some of the diagrams in the paper don't have good consistency when zoomed in.

3.The technology is low-innovative, and multi-view concerns are common in the 3D AIGC field[1].

[1] Liu J, Huang X, Huang T, et al. A comprehensive survey on 3D content generation[J]. arXiv preprint arXiv:2402.01166, 2024.

**Questions:**

See weakness.
I will check the author's response and revise the score after combining other review comments.

---

> ### Author Response · Authors · 2024-11-25
> **Reply to Reviewer EySz Part1**
>
> We thank the reviewer for their insightful comments and suggestions. We address each of the points mentioned in the review below:
>
> > **Re: Weakness1 “Poor qualitative results. I feel that the overfitting is evident in the effect”**
>
> (Please also see point 1&2 in our reply to all reviewers)
>
> To address these concerns we have greatly expanded and improved our OOD evaluations for the text-conditional setting and show many more qualitative examples now for both settings on our updated websites. We have also trained a version of VideoPanda using the more powerful base model CogVideoX-2B and have found greatly improved generalization and visual quality.
>
> **Video condition:** We would like to point out that in the video-conditional setting, our model demonstrates a stronger degree of generalization (beyond the WEB360 domain) as we have demonstrated with evaluation on the out of distribution videos. To further support this, we have expanded our OOD evaluation for the video-conditional setting and show many more examples on our updated website.
>
> Please check some qualitative results from our OOD Video-cond AI generated videos https://mvpanovideo.github.io/VideoPanda/release_col_cond_comp/release_col_cond_comp/index.html
>
> These OOD evaluation videos were sourced from generated samples from SORA/Runway/Luma as downloaded from their respective websites. In total there are 50 videos used for the evaluation. We will add these additional details regarding the selection process of the evaluation set in the supplementary materials.
>
>
> **Text condition:**
>
> Our text-conditional OOD evaluations are now using the full VBench suite of prompts (1700+ prompts) with great diversity and coverage of different dimensions and categories. In the text-conditional case, bad results for OOD prompts are caused in part by our base model not understanding the prompt. We found that switching to a stronger base model (CogVideoX-2B) simply fixes it.
>
> Please check some qualitative results from our OOD Text-cond VBench prompts: https://mvpanovideo.github.io/VideoPanda/release_text_cond_comp/release_text_cond_comp/index.html
>
> We have also computed quantitative metrics (Clip Score, FID and FVD to HDVila videos) for our OOD text-conditional evaluation on VBench prompts. Please see our comment to all reviewers “**7 OOD Evaluation: Quantitative Metrics**”.
>
>
> > **Re: Weakness1 “The watermarks are being generated, and on closer inspection you can see the airpano.”**
>
> (Please see point 6 in our reply to all reviewers)
>
> It is not necessarily the case that generating a watermark indicates that such samples can’t be different from WEB360. For example, quite a few of our OOD text-conditional samples contain the watermark despite being drastically different from content present in WEB360.
> Just to name some: “0) In a charming Parisian café, a panda sits…” and “1) A joyful Corgi with a fluffy coat and expressive eyes” that can be seen here: https://mvpanovideo.github.io/VideoPanda/release_text_cond_comp/release_text_cond_comp/index.html
>
> Note that there are also some examples of our model **not** generating the watermark such as: “(7) A vibrant individual, dressed in a colorful outfit with a..” and left column of *8) A serene individual, dressed in a flowing white shirt and dark trousers” and “(19) A focused individual, wearing a dark apron over a white shirt, stands”. The results look reasonable in these cases.
>
> This could be solved by careful data filtering, or some other techniques (like the DomainAdapter of AnimateDiff [Guo et al., 2023]) to remove the watermark. But this is not the main focus of this paper.
>
>
> > **Re: Weakness2 “Some of the diagrams in the paper don't have good consistency when zoomed in.”**
>
> We thank the reviewer for bringing up this concern. Are you referring to the panorama results?
>
> Part of the seams between view boundaries were a result of our suboptimal stitching code for forming the panorama from the multiple views, we adopt the feathering approach used in the MVDiffusion implementation that smoothly blends images based on distance to image center which removes many of the more subtle seamlines that were apparent before.
>
> In some OOD cases, the model could still exhibit some less subtle inconsistencies between views but we show that these reduce when using a stronger base model (CogVideoX-2B).
> Please refer to our new and greatly expanded OOD evaluation video samples in both the video-conditional and text-conditional cases:
>
> OOD Video-cond AI generated videos: https://mvpanovideo.github.io/VideoPanda/release_col_cond_comp/release_col_cond_comp/index.html
>
> OOD Text-cond VBench prompts: https://mvpanovideo.github.io/VideoPanda/release_text_cond_comp/release_text_cond_comp/index.html

---

> > ### Author Response · Authors · 2024-11-25
> > **Reply to Reviewer EySz Part2**
> >
> > > **Re: Weakness3 “The technology is low-innovative, and multi-view concerns are common in the 3D AIGC field[1].”**
> >
> > (Please also see point 5 in our reply to all reviewers)
> >
> > While, we agree that there are many works in the 3D AIGC field tackling multi-view generation, we are the first to tackle the video-conditional panorama video generation task and applying the multiview-attention mechanisms to improve video panorama generation. We believe the capability of a model to synthesize multiple views of a dynamic scene is key to address many important problems in vision, graphics and robotics. To support this goal, panoramic videos are a very promising yet underutilized source of true multiview spatiotemporal data that contains dynamic objects.
> >
> > We would also like to highlight our multi-task training strategy for letting the same model handle flexible conditioning (see point 5c) and the randomized matrix strategy that shows we can greatly improve the quality of test time generalization to longer sequences with less computational demands (see point 5d).

---

> > ### Comment · Reviewer_EySz · 2024-11-26
> > **Thanks to the author for the reply**
> >
> > After seeing the updated methods and examples again, I raised my rating.

---

> > > ### Author Response · Authors · 2024-11-26
> > > **Thank you for the quick reply!**
> > >
> > > We thank the reviewer for their quick reply and checking our updated results and evaluations!
> > >
> > > Should you have any other concerns or questions, please feel free to let us know and we will be happy to actively address them!

---

### Official Review · Reviewer_cZZr · 2024-11-03

**Soundness:** 3
**Presentation:** 3
**Contribution:** 2
**Rating:** 5
**Confidence:** 3

**Summary:**

This paper presents a novel method for long panoramic video generation from a text prompt or a perspective video. Different from existing 360-DVD that generates equi-rectangular panorama video directly, it builds on existing video diffusion models by adding multi-view attention layers to generate consistent multi-view outputs. It is expect to help maintain the capability of the pre-trained model without domain gap hindering. Extensive experimental results tells the superiority of the proposed method, especially the quantitative evaluation and user study. Anyhow, it still suffers from obvious visual artifacts, as shown in the demonstrated video results.

**Strengths:**

- This paper is well written and easy to follow. Every design is well-motivated and clear clarified.

- The key idea to formulate panorama video generation as multi-view video generation makes sense and novel. Quantiative results (i.e. Table 1) evidence the superiority over existing baseline method 360DVD.

- The experimental evaluation is well-conducted and sufficient. Multiple metrics are adopted and user study is performed.

**Weaknesses:**

-The major weakness is that most of the generated panorama videos are not good as expected, which renders the importance of the key technical innovation not well supported.
First, most of the results present ambiguous semanic structure or broken scene, such as the [autoregressive generation] showcase "anime girl standing on a boat", and the [video-conditioned with different prompts] showcase "A view of the seashore with the sea coming up against a cliff.“ This is a unignorable weakness about the performance.
Besides, almost all results present obvious seamline across views.
It seems the newly introduced multi-view attention does not work as expected. A possible attempt is jointly finetuning the base model with LORA, which may help the model better adapt to the panorama video distribution.

-As stated above, some of results suffers from broken content and semantic ambiguity. In contrast, although showing over-smooth textures, the semantics of 360DVD are more natural and the scene of content are more identifiable than the proposed method. This impedes the justification of the proposed method is better than existing 360DVD.

-The comparative examples with 360DVD are almost static scene, which makes the evaluation less convincing. It is highly required to evaluate on cases with moving objects (such as "moving car on the street", "astronaut riding a horse on grass", etc.), because the consistency of dynamic objects is one of the major focus in video generation task.

**Questions:**

- How many samples are used for the evaluation in Table 2? How do you collected the prompts? Do they cover good diversity?

- It is hard to understand why multi-task training is better than single task training. This is contrast with some common feeling, for example all the T2V or I2V modes are processed with fixed frames, because varied number of tokens in the attention computation may hinder the performance on a specific frames and resolution. I would like to see more elaboration over this.

- The artifacts of full matrix shown in Figure. 7, actually also happen in the result of the proposed method (i.e. random matrix), such as in the [video-conditioned generation] showcase "A view of a panda's face peeking from behind a tree branch amidst lush green foliage“.
So it seems both setting suffers from the over blurry and structure mixture artifacts.

- According to the paper, the model is finetuned with the base model layers frozen. In my understanding, the distribution of each view of the panorama still deviate with the original real-world image. Would it be helpful to the generation quality (such as the ambiguous/broken scene semantics) if the base model is tuned with LORA?

---

> ### Author Response · Authors · 2024-11-22
> **Reply to Reviewer 2**
>
> We thank the reviewer for their insightful comments and suggestions. We address each of the points mentioned in the review below:
>
>
> > **Re: Weakness1 “First, most of the results present ambiguous semanic structure or broken scene, such as the [autoregressive generation] showcase "anime girl standing on a boat", and the [video-conditioned with different prompts] showcase "A view of the seashore with the sea coming up against a cliff.”**
>
> (Please also see point 1,2 and 4 in our reply to all reviewers)
>
> **Video-conditional case**: our model does sometimes exhibit imperfect semantic structure. These cases are generally heavily out of distribution examples and it performs better in more typical cases. We have added many more qualitative examples of our OOD video-conditional generation to our new website page here: https://mvpanovideo.github.io/VideoPanda/release_col_cond_comp/release_col_cond_comp/index.html
>
> We note that this is a data issue rather than a fundamental issue in our model.
>
> We have trained our same model with more panorama video data we collect that contain more varied elevations whereas WEB360 is heavily biased towards zero elevation. The additional data helps for videos with non-zero elevations as can be seen when comparing the left and middle column of (7) “A view of the coastal town, the historic stone structure with a dome, the terraced buildings, and the vast blue sea.” and (8) “A view of a rugged off-road vehicle driving.” show extreme examples of non-standard elevation.
>
> Further, we found that using a more powerful base model, CogVideoX-2B can greatly improve the handling of OOD content which we show on the same website in the rightmost column: VideoPanda (CogVideoX-2B + NewData)
> In particular (5) “A Japanese animated film of a young woman standing on a ship and looking back at camera“ features an extreme closeup of the face. The new model based on CogVideo understands the concept of placing the scene on top of a boat better.
>
> **Text-conditional case**: We would also like to highlight that we have greatly expanded our text-conditional OOD evaluation and use all VBench prompts, presenting many qualitative results here: https://mvpanovideo.github.io/VideoPanda/release_text_cond_comp/release_text_cond_comp/index.html
>
> Despite not being perfect, our model generally creates scenes that are proper panoramas whereas 360DVD features heavy distortions in sky and ground views. Our model is also generally better aligned to the prompt. We corroborate these qualitative findings with our new quantitative evaluations on these OOD cases where we perform much better on Clip Score and FID/FVD to HDVila videos. Please see comment **(7 OOD Evaluation: Quantitative Metrics)** to all viewers above for specifics.
>
>
> > **Re: Weakness1 “Besides, almost all results present obvious seamline across views. It seems the newly introduced multi-view attention does not work as expected.”**
>
> Part of the seams were a result of our suboptimal stitching code for forming the panorama from the multiple views, we adopt the feathering approach used in the MVDiffusion implementation that smoothly blends images based on distance to image center which removes many of the more subtle seamlines that were apparent before.
>
> In some OOD cases, the model could still exhibit some less subtle inconsistencies between views but we show that these reduce when using a stronger base model (CogVideoX-2B).
> Please refer to our new and greatly expanded OOD evaluation video samples in both the video-conditional and text-conditional cases:
>
> OOD Video-cond AI generated videos: https://mvpanovideo.github.io/VideoPanda/release_col_cond_comp/release_col_cond_comp/index.html
>
> OOD Text-cond VBench prompts: https://mvpanovideo.github.io/VideoPanda/release_text_cond_comp/release_text_cond_comp/index.html
>
>
> >**Re: Weakness1 and Question4 “A possible attempt is jointly finetuning the base model with LORA, which may help the model better adapt to the panorama video distribution.” “Would it be helpful to the generation quality (such as the ambiguous/broken scene semantics) if the base model is tuned with LORA?”**
>
> Thank you for the great suggestion. We note that we are currently training our CogVideo model without any freezing. We found that the more powerful base model is better able to preserve its general knowledge even without needing freezing. However, it could be further improved by using a combination of freezing and LoRA as you suggested We are running this experiment now and excited to report on it as soon as we have ablated this. Thanks again for the suggestion!

---

> > ### Comment · Reviewer_cZZr · 2024-11-26
> >
> > Thank you for the carefully prepared results and responses. Through using a more powerful base model, CogVideoX-2B, and leveraging additional panorama videos for training, the updated results have shown significant improvements over the previous ones. However, my concerns are only partially addressed.
> >
> > On the one hand, the proposed method indeed generates visually promising results when the base model and training data are enhanced. On the other hand, the improvements and advantages of the proposed method are largely attributed to the capabilities of the base model and the larger training dataset, rather than the method itself. To me, compared to the baseline 360DVD, the advantage of the proposed method is still unclear. This is because 360DVD is based on a base model like AnimateDiff and uses the WEB360 training dataset. What would happen if these two enhancements were applied to 360DVD? I still need more justification for the superiority of the proposed method over existing baselines.

---

> > > ### Author Response · Authors · 2024-11-26
> > > **Clarifying our comparison with 360DVD using fair data and model setting. Reply to: Official Comment by Reviewer cZZr**
> > >
> > > We thank the reviewer for their quick reply and kind comments about our updated results and presentation! We very much appreciate your constructive feedback (such as suggesting non-paired FID/FVD for evaluation) which improved the quality of our rebuttal.
> > >
> > >
> > > We would like to clarify a few points. In our expanded OOD evaluations, we have made a fair comparison (in terms of data and base model) to 360DVD using the **original model from our paper**.
> > >
> > > All of our quantitative and qualitative comparisons against 360DVD include our VideoPanda model based on **VideoLDM using the same data (WEB360)** which is a fair base model comparison. VideoLDM is based on SD1.5 and shares a very similar structure to AnimateDiff (the base model of 360DVD).
> > >
> > > **For quantitative comparisons:**
> > >
> > > We conducted quantitative OOD text-conditional evaluation where our base model using VideoLDM strongly outperforms 360DVD in Clip Score, FID and FVD (to HDVila3k). We have copied the quantitative comparison here for your convenience:
> > > | Text conditional  Vbench All Dimensions (946 prompts x 3 seeds) | Elevation=+/-60degree 8 Views |                         |             | Horizontal 8 Views (elevation=0) |                         |             |
> > > |-----------------------------------------------------------------|:-----------------------------:|:-----------------------:|:-----------:|:--------------------------------:|:-----------------------:|:-----------:|
> > > |                                                                 | FID (to HDVila3k frames)      | FVD (to HDVila3k video) | Clip Score  | FID (to HDVila3k frames)         | FVD (to HDVila3k video) | Clip Score  |
> > > | 360DVD (AnimateDiff + web360)                                          | 149.7                         | 901.1                   | 23.39       | 127.1                            | 801.9                   | 27.63       |
> > > | Ours (VideoLDM + web360)                                        | **130.6**                     | **826.6**               | **24.12**   | **112.2**                        | **677.8**               | **27.78**   |
> > >
> > > These additional comparisons corroborate our findings in the paper on the in-distribution text-conditional setting (Table 1) where our method also greatly outperforms 360DVD.
> > >
> > >
> > > **For qualitative comparisons:**  (https://mvpanovideo.github.io/VideoPanda/release_text_cond_comp/release_text_cond_comp/index.html  )
> > >
> > >
> > > The left column is our VideoPanda with (VideoLDM+WEB360) and the right column is 360DVD (AnimateDiff+WEB360).
> > >
> > > **Multiview perspective generation VS direct equirectangular generation:**
> > > One main benefit of using multiview perspective generation compared to the equirectangular format used by 360dvd, is that we can more naturally generate content in the sky and ground views (above or below +/-60 degree elevation) that typically are highly distorted in the equirectangular format. This is very visible in the OOD text-conditional samples from 360DVD when viewing the sky or ground views using the provided VR viewer. We also showed this in the appendix contained in the supplementary material (please see Appendix Figure A5).
> > >
> > > Viewing results in VR also highlights the strange motions in 360DVD including the ground morphing in a spiral direction (0) “In a charming Parisian café, a panda”
> > > or inanimate objects sliding in different directions (17) “A serene scene unfolds with a delicate porcelain teacup”.
> > > Such errors are hard to visually spot from only looking at the equirectangular projection native to 360DVD but very important for the actual task of panorama video generation.
> > >
> > >
> > > **Superior range of capability:**
> > >
> > > Additionally, 360DVD only considers the text-conditional case and doesn't consider video-conditional or autoregressive generation at all. We can handle all cases with the same base model. I think this can also be seen as a superiority in the range of capability.
> > >
> > >
> > > We would like to reiterate that our contributions are introducing the video-conditional panorama generation. Our multi-task training strategy enables a unified model capable of flexible conditioning during inference  (see point 5c) and the randomized matrix strategy improves the quality of test time generalization to more frames with less computational demands in training (see point 5d).
> > >
> > >
> > > Please also note that for all of the text-conditional comparison to 360DVD **we did not increase the training data**, but kept it the same (WEB360) for fair comparison. Specifically, both the left column VideoPanda with (VideoLDM+WEB360) and middle column VideoPanda with (CogVideoX-2B+WEB360) are using WEB360 dataset.
> > >
> > >
> > > Again we are very very thankful for your prompt and valuable feedback and please let us know if you still have concerns and we will be happy to actively address them!

---

> ### Author Response · Authors · 2024-11-22
> **Reply to Reviewer cZZr Part 2**
>
> > **Re: Weakness2 “In contrast, although showing over-smooth textures, the semantics of 360DVD are more natural and the scene of content are more identifiable than the proposed method.”**
>
> (From point 3 in our reply to all reviewers)
>
> We hope our new text-conditional OOD evaluations can help address some of the concerns about how our model compares to 360DVD. We evaluate both methods head-to-head on the VBench prompts. We want to point out that the 360DVD results are almost always not generating proper panorama videos. Additionally, we found that the prompt alignment and scene consistency for VideoPanda can all improve simply by applying VideoPanda to a more powerful base model (CogVdeoX-2B).
>
> One main benefit of using multiview perspective generation compared to the equirectangular format used by 360dvd, is that we can more naturally generate content in the sky and ground views (above or below +/-60 degree elevation) that typically are highly distorted in the equirectangular format. We show this in the appendix contained in the supplementary material (please see Appendix Figure A5). It is also very visible in the OOD text-conditional samples when viewing the sky or ground views using the provided VR viewer: https://mvpanovideo.github.io/VideoPanda/release_text_cond_comp/release_text_cond_comp/index.html
>
> Such errors are hard to visually spot from the equirectangular projection native to 360DVD but very important for the actual task of panorama video generation.
> Please have a try!
>
> This finding is also supported by our quantitative evaluation on the OOD prompts.
> Please see the comment **(7 OOD Evaluation: Quantitative Metrics)** to all reviewers for more details regarding the evaluation.
> We present the results in the table below:
> | Text conditional  Input vid:  Vbench All Dimensions (946 prompts x 3 seeds) | Elevation=+/-60degree 8 Views |                          |             | Horizontal 8 View (elevation=0) |                              |             |
> |-------------------------------------------------------------------|:-----------------------------:|:------------------------:|:-----------:|:-------------------------------:|:----------------------------:|:-----------:|
> |                                                                   | FID (to MS-COCO3k)     | FVD (to HDVila3k) | Clip Score  | FID (to MS-COCO3k)     | FVD (to HDVila3k) | Clip Score  |
> | 360DVD                                                            | 128.6                         | 901.1                    | 23.39       | **91.8**                        | 801.9                        | 27.63       |
> | Ours (VideoLDM + web360)                                          | **115.0**                     | **826.6**                | **24.12**   | 92.6                            | **677.8**                    | **27.78**   |
> | Ours (Cogvideo + web360)                                          | **_93.4_**                    | **_675.9_**              | **_25.99_** | **_74.5_**                      | **_624.7_**                  | **_29.33_** |
>
> Note that the original model we presented in our paper (VideoPanda using VideoVLDM base model) is clearly better than 360DVD on all metrics. VideoPanda significantly outperforms 360DVD on elevated views, highlighting its superior ability to generate the ground and sky views, which are distorted in the equirectangular representation used by 360DVD.
> Additionally, the superior performance gained by using the CogVideoX-2B base model with VideoPanda is supported by improvement across all metrics.
>
> **> Re: Weakness3 “The comparative examples with 360DVD are almost static scene.” “It is highly required to evaluate on cases with moving objects (such as "moving car on the street", "astronaut riding a horse on grass", etc.)”**
>
> (Please also see point 1 and 3 in our reply to all reviewers)
>
> We have now included the entire VBench suite for our OOD text-conditional experiments and it contains many prompts with dynamic objects. Please see the qualitative comparisons here:
> https://mvpanovideo.github.io/VideoPanda/release_text_cond_comp/release_text_cond_comp/index.html.
> We do find that the generations from our model with the VideoLDM base model tends to generate videos with lower amount of motion. However, this is related to the base model and VideoPanda on CogVideoX-2B trained on the same data is able to generate scenes with fairly dynamic content.
>
> As 360DVD conditions on optical flow, it can directly force some motions in the scene but the motions can often be incoherent such as objects morphing in and out of existence.
> Viewing results in VR also highlights other strange motions in 360DVD including the ground morphing in a spiral direction (0) “In a charming Parisian café, a panda”
> or inanimate objects sliding in different directions (17) “A serene scene unfolds with a delicate porcelain teacup” when comparing VideoPanda (left column) vs 360DVD (right column).

---

> ### Author Response · Authors · 2024-11-25
> **Reply to Reviewer cZZr Part 3**
>
> > **Re: Question1 “How many samples are used for the evaluation in Table 2? How do you collect the prompts? Do they cover good diversity?”**
>
> Table2 is our comparison to MVDiffusion on the in-distribution video-conditional setting. Table1 is our comparison to 360DVD on in-distribution text-conditional setting. Both of these in-distribution evaluations used the same data source.
> - The number of ground truth panorama videos used for in-distribution evaluation is 100.
> - These videos were selected from online panorama videos. To ensure, the selected videos do not overlap with WEB360, additional videos we sourced from airpano were limited to ones uploaded beyond the creation date of WEB360 and from another channel, “National Geographic”.
> - The prompts for the clips were obtained from captioning the input view extracted from the panorama with CogVLM.
> - In terms of diversity, the clips are similar in distribution to the WEB360 data.
> - We sampled one seed per prompt for doing the evaluations in the paper but for the new OOD evaluations we sample 3 seeds for each method.
>
> We have greatly improved our OOD evaluations including quantitative metrics and using VBench prompts that cover a good diversity of different dimensions and categories.
> Please see the comment **(7 OOD Evaluation: Quantitative Metrics)** to all reviewers for more details regarding the OOD evaluations.
>
>
> **> Re: Question2 “It is hard to understand why multi-task training is better than single task training. This is contrast with some common feeling, for example all the T2V or I2V modes are processed with fixed frames, because varied number of tokens in the attention computation may hinder the performance on a specific frames and resolution. I would like to see more elaboration over this.”**
>
> (From points 5C & 5D in our response to all reviewers)
>
> We want to clarify that in the paper the “multi-task training” refers to training the same model to be capable of handling different types of conditioning (text-, video- and autoregressive). This model is more capable than a model only trained for one of the settings. Performance-wise the multi-task model is on par with the single task video-conditional model albeit slightly worse as shown in Table 3 of the paper. Please also see point 5C in our response to all reviewers for more detailed explanation.
>
> The use of non-fixed frames is the “random matrix” strategy we present in the paper. It is a computational technique to improve the model’s ability to generate full multiview videos of longer time horizons at test time for a given computational budget during training.
>
> Our random matrix strategy is explained in line 476 in the main paper. We wish to generate 8 views with 16 frames but because we are generating more views than single view training, it can not fit in GPU memory and also slows down training. We could still train with 8 views and 6 frames instead which does fit in memory. We call this strategy “fixed matrix”. Although simple, the mismatch in training frames (6) vs inference number of frames (16) results in blurry outputs which we show in the top row of Figure 7. We apologize for an oversight in the caption of Figure 7. It should say “fixed matrix” rather than “full matrix”.
>
> Instead of the reduced fixed matrix training, we notice that one can still fit 3 views at 16 frames. By randomizing which 3 out of the 8 views we choose and randomizing other combinations that can fit in memory such as 4 views at 12 frames, we can strengthen the models ability to generalize to sampling more frames at test time. This strategy is called “random matrix” and its improvement is illustrated in the bottom row of Figure 7. We quantitatively evaluated this strategy on the in-distribution video-conditional task in Table 3, where all quality metrics improve quite a bit but reconstruction PSNR is slightly lower which could be related to slightly better global color consistency for the fixed matrix model that is sees more training iterations featuring all views.
>
> Note that this technique is still applicable for extending the horizon even with larger compute and memory budget (for example even if we use Context Parallel to train with more frames, it doesn't make random matrix useless as it can still improve the performance when the test time temporal window is extended even longer for the same compute cost).
>
>
> > **Re: Question3 “The artifacts of full matrix shown in Figure. 7, actually also happen in the result of the proposed method (i.e. random matrix), such as in .. "A view of a panda's face ...“.”**
>
> Although we do see that oversmoothing could happen for more OOD results, the random matrix strategy largely reduces these cases. We quantitatively evaluated this through ablating random matrix in the paper. Please see the results we present in Table 3 where the random matrix strategy is the middle row and greatly reduces the FVD and FID score compared to the last row which is not using it.

---

### Official Review · Reviewer_TXRj · 2024-11-04

**Soundness:** 2
**Presentation:** 2
**Contribution:** 2
**Rating:** 3
**Confidence:** 5

**Summary:**

This paper introduces a method named VideoPanda, designed to synthesize 360-degree videos from text prompts or single-view videos. VideoPanda builds on existing video diffusion models by adding multi-view attention layers to produce consistent multi-view outputs. Both quantitative and qualitative results are presented.

**Strengths:**

The concept of expanding single-view videos into panoramic videos is interesting. Some of the generated results appear visually good.

**Weaknesses:**

1. The model appears to be overfitted to the WEB360 dataset. The test videos are directly sourced from the WEB360 dataset or selected from similar scenes on the airpano channel (e.g., the pandas in "100329" from the WEB360 dataset, and the ice and mountains in "100666"). The results generated by the method often contain the airpano watermark, whereas the few results without watermarks on the webpage exhibit strange artifacts in other regions. This indicates a lack of generalization to real-world scenarios.
2. The authors state, "Since the out-of-distribution condition inputs do not originate from 360 videos, we cannot compute metrics that require ground truth images, such as pairwise FVD." However, to the best of my knowledge, FID and FVD do not necessarily require paired prediction-ground truth data. FID and FVD are distribution-matching metrics and do not require one-to-one correspondence between generated and real data.
3. There are no ablation studies on the multi-view attention mechanism. The paper does not clearly explain the differences between the proposed multi-view attention and existing methods, such as MVDream.
4. The paper lacks experiments on conditions with varying fields of view (FOV) and view directions. The results only demonstrate conditioning using a 0-degree latitude image. In practical scenarios, adapting to different FOVs and viewing angles is a common requirement.

**Questions:**

1. Can VideoPanda generate videos conditioned on different FOVs and view directions?
2. What is the specific implementation of the multi-view attention, and how does it differ from existing methods?
3. Can the authors provide more quantitative and qualitative results without watermarks?

---

> ### Author Response · Authors · 2024-11-22
> **Reply to Reviewer TXRj Part 1**
>
> We thank the reviewer for their insightful comments and suggestions such as computing non-paired FVD. We address each of the points mentioned in the review below:
>
> **> Re: Weakness1 “The model appears to be overfitted to the WEB360 dataset.”**
>
> (Please also see point 1&2 in our reply to all reviewers)
>
> To address these concerns we have greatly expanded and improved our OOD evaluations for the text-conditional setting and show many more qualitative examples now for both settings on our updated websites.  We have also trained a version of VideoPanda using the more powerful base model CogVideoX-2B and have found greatly improved generalization and visual quality.
>
>
> **Video condition:** We would like to point out that in the video-conditional setting, our model demonstrates a stronger degree of generalization (beyond the WEB360 domain) as we have demonstrated with evaluation on the out of distribution videos. To further support this, we have expanded our OOD evaluation for the video-conditional setting and show many more examples on our updated website.
>
> Please check some qualitative results from our OOD Video-cond AI generated videos: https://mvpanovideo.github.io/VideoPanda/release_col_cond_comp/release_col_cond_comp/index.html
>
> These OOD evaluation videos were sourced from generated samples from SORA/Runway/Luma as downloaded from their respective websites. In total there are 50 videos used for the evaluation. We will add these additional details of the evaluation set in the supplementary materials.
>
>
> **Text condition:**
>
> Our text-conditional OOD evaluations are now using the full VBench suite of prompts (1700+ prompts) with great diversity and coverage of different dimensions and categories. In the text-conditional case, bad results for OOD prompts are caused in part by our base model not understanding the prompt. We found that switching to a stronger base model (CogVideoX-2B) simply fixes it.
>
> Please check some qualitative results from our OOD Text-cond VBench prompts: https://mvpanovideo.github.io/VideoPanda/release_text_cond_comp/release_text_cond_comp/index.html
>
> We have also computed quantitative metrics (Clip Score, FID and FVD to HDVila videos) for our OOD text-conditional evaluation on VBench prompts. Please see our comment to all reviewers “**7 OOD Evaluation: Quantitative Metrics**”.
>
>
> **> Re: Weakness1 “The test videos are directly sourced from the WEB360 dataset or selected from similar scenes on the airpano channel”**
>
> We want to explicitly state that the test videos we used for our in-distribution evaluations were **NOT** sourced from WEB360. To ensure, the selected videos do not overlap with WEB360, additional videos we sourced from airpano were limited to ones uploaded beyond the creation date of WEB360 and from another channel, “National Geographic”. For example the panda video you refer to on the website is from nat geo: https://www.youtube.com/watch?v=0XrH2WO1Mzs. Although they have related content type, the video is different.
>
> As for the ice and mountains (in "100666"), this is from the in distribution text-conditional evaluation. As we mentioned in the paper (line 354), the in distribution text-cond evaluation uses the captions of the in-distribution evaluation videos. In this case, this text prompt corresponds to a caption generated from a clip in this “National Geographic” video: https://www.youtube.com/watch?v=XPhmpfiWEEw&pp=sAQA.
> We agree that for the in-distribution evaluation, there can be similarities between the collected videos and videos in WEB360 which could result in similar text prompts to ones in WEB360 in some cases and we therefore supplemented these with our OOD evaluations.
>
> Please kindly refer to our OOD text-condition and OOD video input evaluations above.
> OOD Text-cond VBench prompts: https://mvpanovideo.github.io/VideoPanda/release_text_cond_comp/release_text_cond_comp/index.html
> OOD Col-cond AI generated videos: https://mvpanovideo.github.io/VideoPanda/release_col_cond_comp/release_col_cond_comp/index.html

---

> ### Author Response · Authors · 2024-11-22
> **Reply to Reviewer TXRj Part 2**
>
> > **Re: Weakness1 and Question 3 “Results generated by the method often contain the airpano watermark, whereas the few results without watermarks on the webpage exhibit strange artifacts in other regions.” “Can the authors provide more quantitative and qualitative results without watermarks?”**
>
> (Please see point 6 in our reply to all reviewers)
>
> It is not necessarily the case that generating a watermark indicates that such samples can’t be different from WEB360. For example, quite a few of our OOD text-conditional samples contain the watermark despite being drastically different from content present in WEB360.
> Just to name some: “0) In a charming Parisian café, a panda sits…” and “1) A joyful Corgi with a fluffy coat and expressive eyes” that can be seen here: https://mvpanovideo.github.io/VideoPanda/release_text_cond_comp/release_text_cond_comp/index.html
>
> Note that there are also some examples of our model **not** generating the watermark such as: “(7) A vibrant individual, dressed in a colorful outfit with a..” and left column of *8) A serene individual, dressed in a flowing white shirt and dark trousers” and “(19) A focused individual, wearing a dark apron over a white shirt, stands”. The results look reasonable in these cases.
>
> This could be solved by careful data filtering, or some other techniques (like the DomainAdapter of AnimateDiff [Guo et al., 2023]) to remove the watermark. But this is not the main focus of this paper.
>
>
> > **Re: Weakness2 “FID and FVD do not necessarily require paired prediction-ground truth data”**
>
> (From point 7 in our reply to all reviewers)
>
> We agree that FID/FVD are distributional measures and thank the reviewer for their suggestion. We now compute FID and FVD for the expanded OOD text-conditional evaluations using the VBench prompts. Since these prompts can be fantastical, there are no ground truth video set that accompany them, instead we use the popular image dataset MS-COCO for FID reference set and video dataset HDVila for our FVD reference set.
> Please see the comment **(7 OOD Evaluation: Quantitative Metrics)** to all reviewers for more details regarding the evaluation.
> We present the results in the table below:
> | Text conditional  Vbench All Dimensions (946 prompts x 3 seeds) | Elevation=+/-60degree 8 Views |                         |             | Horizontal 8 Views (elevation=0) |                         |             |
> |-----------------------------------------------------------------|:-----------------------------:|:-----------------------:|:-----------:|:--------------------------------:|:-----------------------:|:-----------:|
> |                                                                 | FID (to HDVila3k frames)      | FVD (to HDVila3k video) | Clip Score  | FID (to HDVila3k frames)         | FVD (to HDVila3k video) | Clip Score  |
> | 360DVD                                                          | 149.7                         | 901.1                   | 23.39       | 127.1                            | 801.9                   | 27.63       |
> | Ours (VideoLDM + web360)                                        | **130.6**                     | **826.6**               | **24.12**   | **112.2**                        | **677.8**               | **27.78**   |
> | Ours (Cogvideo + web360)                                        | **_109.9_**                   | **_675.9_**             | **_25.99_** | **_97.2_**                       | **_624.7_**             | **_29.33_** |
>
> Note that the original model we presented in our paper (VideoPanda using VideoVLDM base model) is better than 360DVD on all metrics.
> VideoPanda significantly outperforms 360DVD in the 60 degree elevation views, highlighting its superior ability to generate the ground and sky views, which are distorted in the equirectangular representation used by 360DVD.
> Additionally, the superior performance gained by using the CogVideoX-2B base model with VideoPanda is clearly seen by the quantitative evaluation above, with massive improvements observed across all metrics.
>
>
> > **Re: Weakness3 and Question3 “There are no ablation studies on the multi-view attention mechanism. The paper does not clearly explain the differences between the proposed multi-view attention and existing methods, such as MVDream.” “There are no ablation studies on the multi-view attention mechanism”**
>
>
> (Please also see point 5B in our reply to all reviewers)
>
> We describe our multi-view attention mechanism in Section 3.1, starting at line 190. There is no inherent difference to multi-view attention mechanisms used by other works such as Cat3D [Gao* et al., 2024] and MVDream [Shi et al., 2023b)]. In the paper, we do not claim the multi-view attention mechanism as a novel contribution. Instead we are the first to utilize it for the generation of panoramic videos and natural handling of input video conditioning.
>
> Please let us know if you would like us to try any specific ablations. Thanks!

---

> ### Author Response · Authors · 2024-11-25
> **Reply to Reviewer TXRj Part 3**
>
> > **Re: Weakness4 and Question1 “The paper lacks experiments on conditions with varying fields of view (FOV) and view directions.”“Can VideoPanda generate videos conditioned on different FOVs and view directions?”**
>
> (From point 4 in our reply to all reviewers)
>
> We agree that handling varying FOV and view elevations is an interesting and important problem for wider practical application.
> Even with our current model, it is able to handle camera variation to some extent due to natural elevation changes in the training videos. Although WEB360 contains very limited elevation changes, the new dataset we collect contains more and we highlight examples of these cases in the video-conditional setting here: https://mvpanovideo.github.io/VideoPanda/release_col_cond_comp/release_col_cond_comp/index.html
>
> The additional data helps the understanding of different elevations as can be seen in the middle column for example (7) “A view of the coastal town, the historic stone structure with a dome, the terraced buildings, and the vast blue sea.” and (8) “A view of a rugged off-road vehicle driving.” show extreme examples of non-standard elevation. Using the CogVideoX-2B as base model also further boosts the visual quality and handling of OOD content.
>
> That being said, handling of other FOV is approximate and extreme elevation changes can also be confusing to our model.
> We also commented on this limitation in the conclusion of the main paper, line 529: To handle such extreme camera variations, we could combine our method with CamFreeDiff [(Yuan et al., 2024)] which has been applied in the image panorama generation setting. We leave this important exploration to future work.

---

> > ### Comment · Reviewer_TXRj · 2024-12-02
> > **Response for Rebuttal**
> >
> > Thank you for the detailed response and the additional experiments provided. The use of a more powerful base model, CogVideoX, has indeed led to some improvements in the quality of panoramic video generation. I have carefully reviewed all reviewers' comments as well as the authors’ responses. However, the following concerns remain unresolved:
> >
> > 1. Limited Novelty: The paper lacks specific innovations tailored to the task of panoramic video generation. As acknowledged in the authors’ response, the multi-view attention mechanism employed is identical to that of MVDream, without any special adaptations for the unique challenges of panoramic video generation. This limitation is reflected in the additional experiments, where noticeable multi-view stitching artifacts are still evident in VR previews, even when using the stronger CogVideoX model.
> >
> > 2. Unfair Comparisons: As pointed out by reviewers U9nk and cZZr, performance improvements derived solely from adding data and switching to a more powerful base model do not constitute a fair comparison. Moreover, the authors have not updated the descriptions or clarified the methodology for FID and FVD metrics in the main text, nor have they incorporated the revised metrics into the manuscript.
> >
> > 3. Response to "Varying FOVs and View Directions": The authors’ reply regarding adaptation to diverse input conditions is unsatisfactory. Accommodating user diversity is crucial for practical applications, especially for tasks such as generating panoramic videos from single-view inputs. Such considerations should be reflected in the methodological design rather than being deferred to future work or partially addressed merely by expanding training data.
> >
> > Given these unresolved concerns, I maintain my initial score.

---

### Author Response · Authors · 2024-11-21
**Summary Response to All (we will also individually reply to each reviewer shortly)**

We thank all the reviewers for their time and great feedback! To address some of the common concerns raised by the reviewers, we made the following additions.


**1 Expanding and Improving OOD Evaluations**

We greatly expanded and improved our OOD evaluations for both the text-conditional and video-conditional settings.

Our text-conditional OOD evaluations are now using the full VBench suite of prompts (1700+ prompts) with great diversity and coverage of different dimensions and categories.

Our video-conditional OOD evaluations are using the 50 videos we curate from publicly available AI generated videos and they can feature cases with non-standard FOV and elevation.

We show many additional qualitative examples now on our website (along with comparison to 360DVD). Please see the qualitative results at links below:

OOD Text-cond VBench prompts: https://mvpanovideo.github.io/VideoPanda/release_text_cond_comp/release_text_cond_comp/index.html

OOD Col-cond AI generated videos: https://mvpanovideo.github.io/VideoPanda/release_col_cond_comp/release_col_cond_comp/index.html

Please take a look!

Update: We have computed quantitative metrics for our OOD evaluation comparing VideoPanda to 360DVD for the text-conditional setting on using all 946 VBench-all_dimension prompts including Clip score, FVD to HDVila3k and FID to HDVila3k first frames. Unpaired FVD and FID was based on suggestion from Reviewer TXRj. We thank them very much for the suggestion. Our original VideoPanda model on VideoLDM (presented in the paper) performs better than 360DVD across all the metrics on this benchmark. Please see the quantitative metrics in the comment **(Point 7 OOD Evaluation: Quantitative Metrics)** above.


**2 Results on new base model (CogVideoX-2B)**

Many reviewers were concerned about the OOD generalization ability of our VideoPanda model.
To clarify, in the main paper we used the VideoLDM model (not SVD) which was presented in the work Align your Latents [Blattmann et al., 2023a] as cited in our paper. This model is a text-to-video model based on SD1.5 and is quite similar in design to AnimateDiff.


It shows quite good generalization in the video-conditional setting but some OOD prompts can be challenging in the text-conditional setting. Please see the extended qualitative OOD video and text-conditional results on the updated websites linked above (in point 1) for a better understanding of the model’s capabilities.
We found that this weakness on OOD text-conditional setting primarily stems from the weakness of the base model itself not understanding the prompts.


So we implemented the VideoPanda method on top of a more powerful model, CogVideoX-2B which is a medium-sized transformer DiT.

Our method readily applies without any major alterations. Per-frame multi-view attention layers are added to the model in between the 3D video attention that now operates on each of the views separately. The weights of the multiview-attention is initialized from the 3D video attention layer and zero-residual initialized.
We finetune the full model end-to-end using the same dataset (WEB360). Additional training details will be included in the supplementary materials.


With no major alterations to our framework, the new model VideoPanda (CogVideoX-2B) demonstrates large improvements in OOD generalization and video quality in both the text-conditional and video-conditional settings.


We have also addressed each reviewer individually with our responses.
Once again we want to thank the reviewers for their great feedback and suggestions.

[Yang et al, 2024] Yang, Zhuoyi et al. “CogVideoX: Text-to-Video Diffusion Models with An Expert Transformer.” ArXiv abs/2408.06072 (2024): n. pag.

---

> ### Author Response · Authors · 2024-11-21
> **General response, continued: Non-standard Camera Elevation Input**
>
> **3 Comparison to 360DVD**
>
> We hope our text-conditional OOD evaluations can help address some of the concerns about how our model compares to 360DVD. We evaluate both methods head-to-head on the VBench prompts. We want to point out that the 360DVD results are almost always not generating proper panorama videos.
>
> One main benefit of using multiview perspective generation compared to the equirectangular format used by 360dvd, is that we can more naturally generate content in the sky and ground views (above or below +/-60 degree elevation) that typically are highly distorted in the equirectangular format. We show this in the appendix contained in the supplementary material (please see Appendix Figure A5). It is also very visible in the OOD text-conditional samples when viewing the sky or ground views using the provided VR viewer: https://mvpanovideo.github.io/VideoPanda/release_text_cond_comp/release_text_cond_comp/index.html
>
> Viewing results this way also highlights the strange motions in 360DVD including the ground morphing in a spiral direction (0) “In a charming Parisian café, a panda”
> or inanimate objects sliding in different directions (17) “A serene scene unfolds with a delicate porcelain teacup” when comparing VideoPanda (left column) vs 360DVD (rightmost column) here: https://mvpanovideo.github.io/VideoPanda/release_text_cond_comp/release_text_cond_comp/index.html.
>
> Such errors are hard to visually spot from the equirectangular projection native to 360DVD but very important for the actual task of panorama video generation.
> Please have a try!
>
>
>
> **4 Video-conditional setting: Handling variations in camera elevation in input video**
>
> Even with our current model, it is able to handle camera variation to some extent due to a few natural elevation changes in the training videos.
>
> However WEB360 contains very limited elevation changes in the training dataset, usually showing smooth landscape fly-overs.
> We hence collect a new dataset of panorama videos and show that it can enhance the models understanding of variations in elevation.
> Furthermore, similarly to the text-conditional setting, we also trained VideoPanda using the more powerful CogVideoX-2B base model which boosts the visual quality and enhances the handling of OOD content further.
> We highlight these new results in the OOD Video-conditional results here (in particular, on the left column is the result from the model in our paper, the middle column is the same model but trained with the new data instead of web360 and the right column is further switching the base model to CogVideoX-2B): https://mvpanovideo.github.io/VideoPanda/release_col_cond_comp/release_col_cond_comp/index.html
>
> The additional data helps the understanding of different elevations as can be seen in for example (7) “A view of the coastal town, the historic stone structure with a dome, the terraced buildings, and the vast blue sea.” and (8) “A view of a rugged off-road vehicle driving.” show extreme examples of non-standard elevation.

---

> ### Author Response · Authors · 2024-11-22
> **General response, continued: 5 Motivation and Novelty**
>
> **5 Motivation and Novelty**
>
> As we mentioned in the paper, our main contributions are as follows:
>
> A: Propose Video-conditional Panorama Video Generation Task
>
> We are the first to introduce the video conditional multiview generation task on open domain. We believe the capability of a model to synthesize multiple views of a dynamic scene is key to address many important problems in vision, graphics and robotics. To support this goal, panoramic videos are a very promising yet underutilized source of true multiview spatiotemporal data that contains dynamic objects.
>
> There are many existing tools for forming image panoramas of static scenes from image collections or videos captured by handheld cameras and phones. These collate and compose the images. However, this approach is not applicable for producing panorama videos of dynamic scenes as the different views will be temporally inconsistent with each other. To capture panoramic videos, customized camera rigs need to be used.
>
> Generating panoramas conditional on input videos is hence an important step to democratize the creation of panoramic content. Such content can be used for creating VR experiences and as backgrounds for visual effects and other graphics applications.
> Another interesting use case is scene reconstruction, where diffusion models have made recent contributions to supply high fidelity priors to resolve inherent ambiguities. Multiview image modeling capabilities for this use case are being enhanced by leveraging pretrained video models but are mostly still restricted to static scene cases.
> Such capabilities may also aid robotic applications where we may wish to playback a recorded sequence but allow the agent to modify its behavior such as viewing direction to some extent.
>
>
> B: Using Multiview-Attention for the Panorama Video Generation
>
> The multi-view attention mechanism we use is not inherently different from existing ones like MVDream or CAT3D, but we introduce it to the video panorama generation setting and show we can get good results whilst keeping the multiview-attention to operate only per-frame and let the rest of the layers keep interpreting each full video separately.
>
> C: Multi-Task Training for Flexible Conditioning
>
> We introduce multi-task training to enable a single model to handle text-conditioning, video-conditioning and autoregression which saves training time and is more convenient. While dropping conditioning during training is a common technique as is used in models like GLIDE, Imagen, and SVD, It is mostly done to enable classifier-free guidance rather than flexible conditioning for a single model.
> The use of joint video and text conditioning in our model is similar to other video outpainting settings and also related to the task of Instruct-Pix2Pix. In our case, our single model is able to handle full generation from text as well as conditional on a video, and autoregression.
>
> D: Randomized Matrix Training Strategy for Longer Temporal Horizon
>
> We introduced the randomized matrix training strategy that improves the model’s ability to generate full multiview videos of longer time horizons at test time for a given computational budget during training.
>
> Our random matrix strategy is explained in line 476 in the main paper. We wish to generate 8 views with 16 frames but because we are generating more views than single view training, it can not fit in GPU memory and also slows down training.  We could still train with 8 views and 6 frames instead which does fit in memory. We call this strategy “fixed matrix”. Although simple, the mismatch in training frames (6) vs inference number of frames (16) results in blurry outputs which we show in the top row of Figure 7. We apologize for an oversight in the caption of Figure 7. It should say “fixed matrix” rather than “full matrix”.
>
> Instead of the reduced fixed matrix training, we notice that one can still fit 3 views at 16 frames. By randomizing which 3 out of the 8 views we choose and randomizing other combinations that can fit in memory such as 4 views at 12 frames, we can strengthen the models ability to generalize to sampling more frames at test time. This strategy is called “random matrix” and its improvement is illustrated in the bottom row of Figure 7. We quantitatively evaluated this strategy on the in-distribution video-conditional task in Table 3, where all quality metrics improve quite a bit but reconstruction PSNR is slightly lower which could be related to slightly better global color consistency for the fixed matrix model that is sees more training iterations featuring all views.
>
> Note that this technique is still applicable for extending the horizon even with larger compute and memory budget (for example even if we use Context Parallel to train with more frames, it doesn't make random matrix useless as it can still improve the performance when the test time temporal window is extended even longer for the same compute cost).

---

> ### Author Response · Authors · 2024-11-22
> **General response, continued: 5 Watermark**
>
> **6 Watermark**
>
> Generation of watermarks are a dataset bias. When training on web360 (almost all the training videos have the airpano watermark), it’s expected that the model will generate the same watermark in inference.
>
> It is not necessarily the case that generating a watermark indicates that such samples can’t be different from WEB360. For example, quite a few of our OOD text-conditional samples contain the watermark despite being drastically different from content present in WEB360.
> Just to name some: “0) In a charming Parisian café, a panda sits…” and “1) A joyful Corgi with a fluffy coat and expressive eyes” that can be seen here: https://mvpanovideo.github.io/VideoPanda/release_text_cond_comp/release_text_cond_comp/index.html
>
> On our newly collected dataset used to support the handling of camera elevation variations (see point 4 above), we notice there are other types of watermarks typically appearing on the bottom views. Also a lot of these videos are shot from a camera held by a person, mounted on their head or mounted on a vehicle of some sort, which is often also adapted by the model. This could be solved by careful data filtering, or some other techniques (like the DomainAdapter of AnimateDiff [Guo et al., 2023]) to remove the watermark. But this is not the main focus of this paper.

---

### Author Response · Authors · 2024-11-25
**Point 7 OOD Evaluation: Quantitative Metrics**

**7 OOD Evaluation: Quantitative Metrics**

As mentioned earlier, we have improved and expanded our OOD evaluations. For the text-conditional setting, we now use the popular VBench evaluation prompts. These prompts offer good diversity and include fantastical examples, such as "pandas sitting in a cafe," to stress test OOD generalization. Qualitative visual results are shown here: https://mvpanovideo.github.io/VideoPanda/release_text_cond_comp/release_text_cond_comp/index.html
To evaluate our performance on OOD cases, we also compute and report some quantitative metrics. These are done for the text-conditional setting, in which we compare 360DVD against VideoPanda using VideoLDM (from the paper) as well as VideoPanda using CogVideoX-2B which we explained further in point 2 above.

We used the gpt enhanced prompts released by VBench here: https://github.com/Vchitect/VBench/tree/master/prompts/gpt_enhanced_prompts

**Clip Score:** We compute a clip score between our generated videos and the input prompt to quantify video-text alignment.

**FID/FVD**: Following the suggestion by reviewer TXRj, we evaluated non-paired FID and FVD for the OOD text-conditional case (meaning that the true set of videos and our generated videos do not share the same prompts as we do not have a reference ground truth set to correspond with the prompts).
For this we decided to use the popular video dataset HDVila for the reference set. In particular, we use 3,000 random videos from HDVila for FVD computations and use the first frames from the same set for FID computations. As this dataset contains perspective view data, we also extract perspective views from our generated panorama videos in order to perform a fair comparison.

**Perspective View Extraction** In addition to the 8 horizontal (zero degree elevation) extracted views we used in the paper, we also included views with non-zero elevation. In particular, we create an additional setting where we extract 8 views in total with 4 views at negative 60 degree elevation looking downwards and another 4 views at positive 60 degree elevation looking upwards. The FOV is kept at 90 degrees. We refer to this setting as “Elevation=+/-60degree Views” in the table below.
These views better capture a complete picture of the panorama while still remaining within the distribution of natural camera angles.

We present the results in the table below:
| Text conditional  Vbench All Dimensions (946 prompts x 3 seeds) | Elevation=+/-60degree 8 Views |                         |             | Horizontal 8 Views (elevation=0) |                         |             |
|-----------------------------------------------------------------|:-----------------------------:|:-----------------------:|:-----------:|:--------------------------------:|:-----------------------:|:-----------:|
|                                                                 | FID (to HDVila3k frames)      | FVD (to HDVila3k video) | Clip Score  | FID (to HDVila3k frames)         | FVD (to HDVila3k video) | Clip Score  |
| 360DVD                                                          | 149.7                         | 901.1                   | 23.39       | 127.1                            | 801.9                   | 27.63       |
| Ours (VideoLDM + web360)                                        | **130.6**                     | **826.6**               | **24.12**   | **112.2**                        | **677.8**               | **27.78**   |
| Ours (Cogvideo + web360)                                        | **_109.9_**                   | **_675.9_**             | **_25.99_** | **_97.2_**                       | **_624.7_**             | **_29.33_** |

The original model we presented in our paper (VideoPanda using VideoVLDM base model) outperforms 360DVD across all of these metrics.
VideoPanda significantly outperforms 360DVD in the 60 degree elevation views, highlighting its superior ability to generate the ground and sky views, which are distorted in the equirectangular representation used by 360DVD.
Additionally, the superior performance gained by using the CogVideoX-2B base model with VideoPanda is clearly seen by the quantitative evaluation above, with massive improvements observed across all metrics.

---

### Meta-Review · Area_Chair_LfHR · 2024-12-19

**Metareview:**

The paper received overall negative scores. The reviewers liked the concept of extending single view videos into panoramic videos. At the same time, they listed a bunch of weaknesses, such as weaker overall results, semantic issues, not convincing evaluation, lack of novel ideas. The only reviewer (EySz) who gave a somewhat positive score, provided a superficial and very short review. Hence the recommendation is to reject the manuscript.

**Additional Comments On Reviewer Discussion:**

There was a very fruitful discussion between authors and reviewers. As far as AC can tell the authors did a good job in trying to address the concerns. And indeed, the scores went up during the discussion period. The authors provided more results with a strong video model. This was appreciated by reviewers. Yet, this was not sufficient, as concerns remained: novelty, comparisons, dependence on the video model and so on.

---

### Decision · Program_Chairs · 2025-01-22

Reject